# An apical Phe-His pair defines the Orai1-coupling site and its occlusion within STIM1

Yandong Zhou [1,6] ✉, Michelle R. Jennette[1,6], Guolin Ma [2], Sarah A. Kazzaz [1], James H. Baraniak[1], Robert M. Nwokonko[3], Mallary L. Groff [1], Marcela Velasquez-Reynel[1], Yun Huang [4], Youjun Wang [5] & Donald L. Gill [1] ✉

Ca[2+] signal-generation through inter-membrane junctional coupling between endoplasmic reticulum (ER) STIM proteins and plasma membrane (PM) Orai channels, remains a vital but undefined mechanism. We identify two unusual overlapping Phe-His aromatic pairs within the STIM1 apical helix, one of which (F394-H398) mediates important control over Orai1-STIM1 coupling. In resting STIM1, this locus is deeply clamped within the folded STIM1-CC1 helices, likely near to the ER surface. The clamped environment in holo-STIM1 is critical—positive charge replacing Phe-394 constitutively unclamps STIM1, mimicking store-depletion, negative charge irreversibly locks the clamped-state. In store-activated, unclamped STIM1, Phe-394 mediates binding to the Orai1 channel, but His-398 is indispensable for transducing STIM1-binding into Orai1 channel-gating, and is spatially aligned with Phe-394 in the exposed Sα2 helical apex. Thus, the Phe-His locus traverses between ER and PM surfaces and is decisive in the two critical STIM1 functions—unclamping to activate STIM1, and conformational-coupling to gate the Orai1 channel.

Cytosolic Ca[2+] signals controlling many essential cellular processes are generated through the coordinated release of Ca[2+] stored within the endoplasmic reticulum (ER) and entry of Ca[2+] across the plasma membrane (PM). This coordination is mediated by ER membrane STIM proteins, which are sensors of luminal Ca[2+] and translocate into ER-PM junctions where they bind to and activate the family of PM Orai Ca[2+] entry channels[1–4]. This process of "store-operated" Ca[2+] entry (SOCE) is essential for generating local Ca[2+] signals and global Ca[2+] oscillations that regulate gene expression and the growth state of most cell types[1–3,5–7]. Human loss- or gain-of-function mutations in STIM or Orai genes result in severe immune, muscular and blood cell dysfunctions[2,8]. Changes in the STIM-Orai signaling pathway play significant roles in the pathophysiology of some major disease states including cancer and cardiovascular, immunological, respiratory, and skin disorders, hence STIM and Orai proteins may be important targets to modify such diseases[8–14]

The fundamental role and broad significance of the SOCE pathway has spurred major advances in understanding the unusual inter-membrane communication mediated by STIM and Orai proteins[1–4,7]. STIM1 is a dimeric protein undergoing extreme molecular and cellular dynamics[1,2,4,7,15–17]. In the resting state, the three CC1 helices of the STIM1 cytoplasmic domain are held within a folded or "clamped" configuration (Supplementary Fig. 1a)[2,7,18–25]. Upon activation by ER Ca[2+] depletion, the two luminal Ca[2+]-sensing domains and the attached single transmembrane (TM) helices, pair together within the STIM1 dimer[19,23,27] (Supplementary Fig. 1). Pairing of the TM helices causes a release from the clamped state. Thereafter, the dimeric STIM1 protein undergoes massive unfolding of the three CC1 helices which

[1]Department of Cellular and Molecular Physiology, The Pennsylvania State University College of Medicine, Hershey, PA 17033, USA. [2]Institute of Biosciences and Technology, College of Medicine, Texas A&M University, Houston, TX 77030, USA. [3]Department of Molecular and Cellular Physiology, Stanford University School of Medicine, Stanford, CA, USA. [4]Center for Epigenetics and Disease Prevention, Institute of Biosciences and Technology, Texas A&M University, Houston, TX 77030, USA. [5]Beijing Key Laboratory of Gene Resources and Molecular Development, College of Life Sciences, Beijing Normal University, Beijing 100875, PR China. [6]These authors contributed equally: Yandong Zhou, Michelle R. Jennette. ✉e-mail: zhouyd@psu.edu; dongill@psu.edu

reassociate into an elongated configuration[20,24–26,28] to expose a critical segment of the protein known as the STIM-Orai activating region (SOAR), the minimal unit of STIM1 that can activate the Orai1 channel[15,29–33]. In its resting state, STIM1 is diffusely distributed throughout the ER cisternae, and SOAR is sequestered within the folded CC1 helices and unavailable for coupling to Orai channels. But, in its activated, unfolded state, STIM1 undergoes a visibly dramatic "punctal" redistribution as it diffuses into and becomes trapped within ER-PM junctions[1,2,34,35]. The unfolded, elongated STIM1 dimer extends approximately 15–20 nm from the ER surface and, upon diffusing into pre-existing ER-PM junctions, its two Lys-rich C-termini attach to acidic lipid headgroups on the PM surface and latch STIM1 to the PM[35,36]. As more STIM1 proteins are ensnared, the junctions enlarge and the exposed SOAR domains are able to tether available PM Orai1 channels by avidly binding to their short cytoplasmic C-terminal M4x peptides[4,7,37–39]. The attachment of SOAR to Orai1 leads to a conformational coupling event in which binding is transduced into gating of the Orai1 channel.

Despite intense study, the nature of the STIM1-Orai1 coupling mechanism to activate the channel remains largely unknown. Thus, the molecular components of STIM1 that not only bind to Orai1 but, in addition, mediate the long-sought conformational coupling that transduces binding into gating of the Orai1 channel, have yet to be identified. Uncertainty also remains in the nature of the clamped resting state of STIM1, the location of clamped SOAR relative to the ER membrane surface, and how unclamping is triggered to allow SOAR release. Structural understanding of SOAR is key to addressing these questions. The human SOAR crystal structure revealed four helices, the two longer helices (Sα1, Sα4) forming a remarkably stable inter-dimer backbone (Supplementary Fig. 1b)[40]. We revealed that the intervening short Sα2 helix forms an exposed "apex" that serves as the critical site for Orai1 interactions[33,37]. Earlier studies utilized expressed SOAR and isolated fragments from STIM1 and Orai1 to derive models on the coupling to Orai1 channels[22,41,42], but there remains uncertainty on such models for SOAR interactions with Orai1 channels[3,7,24,25]. Indeed, the molecular configuration of the SOAR unit is dependent on the complex structure of the *entire* STIM1 protein. Thus, in vivo, SOAR does not function in isolation; in resting STIM1, SOAR is confined within the tightly folded CC1 helices, and in activated STIM1, SOAR is clustered with Orai channels within the highly restricted ER-PM junctional space[7,16,25]. Therefore, key to understanding the complex molecular architecture of STIM1 is analysis of the holo-STIM1 protein. To this, recent elegant single-molecule FRET studies from Lewis's lab[25] have provided new information on the structure and folding of STIM1, revealing that the STIM1 CC1 helices fold in an inter-dimer "domain swapping" configuration around SOAR[43]. These studies confirmed the structure of SOAR and, interestingly, revealed that within the STIM1 protein, the SOAR apical region has some flexibility that could be important for coupling to Orai1.

The current work focuses on the function of the SOAR domain within the STIM1 protein, how it binds to Orai1 channels within the ER-PM junction, and more importantly, how it transduces the Orai1 binding interaction into channel activation. We reveal that these two paradigms can be quite distinct. To obviate the problems and inconsistences of using isolated SOAR or STIM1 fragments, our current studies on SOAR function utilized holo-STIM1 protein constructs; these were expressed in CRISPR-derived STIM1/STIM2 double knockout cells[16] to avoid endogenous STIM protein interference. We reveal that the two amino acids, Phe-394 and His-398 within the small but crucial apical Sα2 helix of STIM1 (Supplementary Fig. 1c), are important in mediating not only Orai1 binding, but also the conformational coupling that mediates Orai1 gating. Thus, while Phe-394 has a decisive role in binding to Orai1, the His-398 residue, while not mediating Orai1 binding per se, is indispensable for "transducing" STIM1 binding into gating of the Orai1 channel. Significantly, the two amino acids are

separated by one helical turn in the Sα2 helix, hence the two adjacent aromatic headgroups can undergo aromatic pairing. Thus, the STIM1-induced transduction of Orai1 binding into channel activation requires a remarkable molecular precision in the distance and nature of the two head groups in the Phe-His pair. Unexpectedly, we determined that the headgroup at position-394 in the apical Sα2 helix, also has a remarkably precise influence on the unfolding of STIM1 in its resting state—positive charge causes store-independent, complete unfolding and translocation of STIM1 into ER-PM junctions—negative charge locks the clamped state and irreversibly blocks any unfolding. These effects indicate the location of the Sα2 apical helix is deep within the folded STIM1, and likely to be close to the ER membrane. The results reflect the extraordinarily critical nature of the immediate environment of the SOAR apex within the clamped resting STIM1 protein, and reveal the precision with which the same SOAR apex undergoes conformational coupling with Orai1 to gate the channel.

## Results

### Roles of two overlapping Phe-His pairs in the apical Orai-coupling site in STIM1

Aromatic pairing within proteins provides important intra- and inter-helical stabilization[44–46]. Phe-His pairs, usually separated by 3 intervening residues, frequently stabilize α-helical domains particularly at the C-terminal ends of short α-helices[45]. We identified two such Phe-His pairs existing within the short sequence forming the strategically exposed "SOAR apex" domain of STIM1 which includes the 7-amino acid, Sα2 helix (Supplementary Fig. 1). This region is shown to be important in the interaction of STIM1 with Orai channels[33,37]. The two Phe-His pairs (named here as inner, F391/H395, and outer, F394/H398; see Supplementary Fig. 1c) exist in an unusual overlapping configuration based on the SOAR crystal structure[40]. Our results here reveal that both Phe-His pairs in STIM1 are involved in mediating the interaction of STIM1 with Orai1, but the outer pair is key to "transducing" the STIM1-Orai1 binding into channel gating. Using CRISPR-derived STIM1/STIM2 double knockout cells stably expressing Orai1-CFP (HEK-S1S2dKO-O1CFP cells) we initially examined replacing and swapping the aromatic residues within the inner Phe-His pair of expressed STIM1-YFP. His replacement of F391 (giving a His-His pair) resulted in loss of STIM1-YFP colocalization with Orai1-CFP (Fig. 1a; Supplementary Fig. 2), as well as substantially reduced STIM1-Orai1 FRET interactions (Fig. 1b, c; Supplementary Fig. 2), and Orai1-mediated $Ca^{2+}$ entry (Fig. 1d, e). This loss of binding and function was at least partially overcome by 2-APB, the powerful SOCE-modifier that likely targets the STIM1-Orai1 coupling process[39,47,48]. Reversed orientation of the inner Phe-His pair with the F391H/H395F double mutation, gave wildtype-like STIM1-Orai1 colocalization, FRET, and $Ca^{2+}$ entry (Fig. 1a–e) suggesting the reversed residues may still allow an aromatic interaction mimicking the wildtype. The H395F mutation giving a Phe-Phe pair also caused little change from wildtype (Fig. 1a, e). The effects of replacing and exchanging aromatic residues in the STIM1 outer Phe-His pair were more striking and informative. Replacement of the F394 residue with His (F394H; giving a His-His pair) caused an almost complete loss of Orai binding (Fig. 1f–h) and Orai1-mediated $Ca^{2+}$ entry (Fig. 1i, j), indicating F394 plays a pivotal role in Orai1 binding. This effect could be transiently reversed by 2-APB. More significantly, mutation of H398 to Phe (H398F, giving a Phe-Phe pair) caused almost no change in STIM1-Orai1 colocalization or FRET interactions (Fig. 1f–h) yet completely blocked store-dependent $Ca^{2+}$ entry (Fig. 1i, j). Even 2-APB could not restore $Ca^{2+}$ entry. Reversal of the Phe-His pair to His-Phe using the double mutation F394H/H398F, resulted in a similar phenotype to the H398F point mutation (Phe-Phe), with full STIM1-Orai1 interactions measured by colocalization or FRET (Fig. 1f–h) but complete loss of any $Ca^{2+}$ entry, and no restoration of either binding or $Ca^{2+}$ entry with 2-APB (Fig. 1i, j).

These results on the function of the outer Phe-His pair of residues have important implications. Changing the F394 residue to a histidine

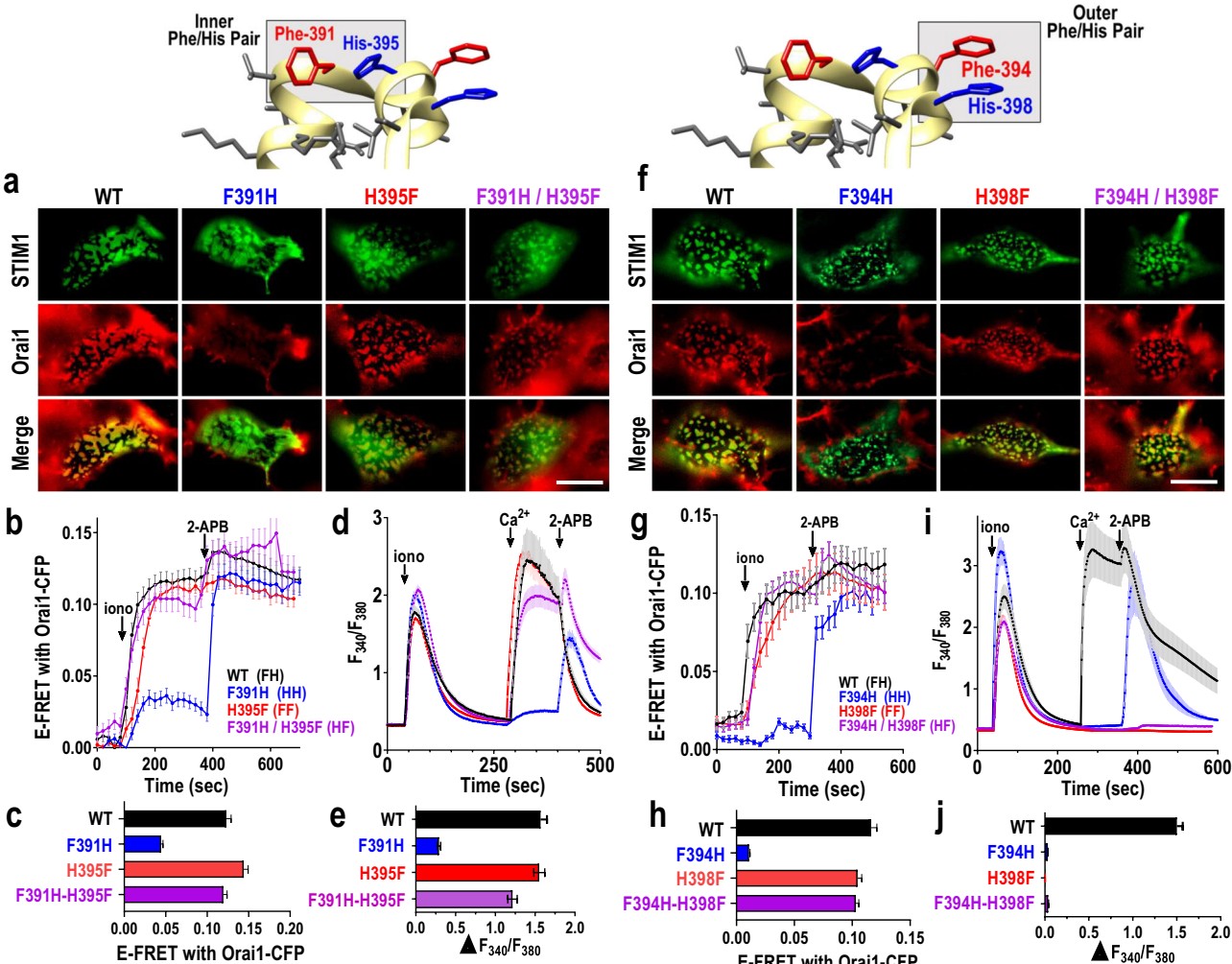

**Fig. 1 | Two Phe-His pairs in the critical SOAR apex STIM1 exert important control of Orai1 binding and channel activation. a–e** Modification of the inner Phe-His pair. **a** Deconvolved images of the ER–PM interface in HEK-S1S2dKO-O1CFP cells transiently expressing STIM1-YFP-WT, or the mutants thereof, F391H, H395F and F391H/H395F. Images of STIM1-YFP (green), Orai1-CFP (red), or merged images were 5 min after store-depletion with 2.5 μM ionomycin. Scale bar = 10 μm. **b** Time course of E-FRET interactions between Orai1-CFP and full-length STIM1-YFP-WT (n = 17), or F391H (n = 19), H395F (n = 21) or F391H/H395F (n = 26) mutants thereof, before and after 2.5 μM ionomycin. **c** Summary data for E-FRET analyses in (**b**), showing the change in E-FRET from baseline (0 s) to the peak after ionomycin addition (350 s). STIM1-YFP-WT (n = 102), F391H (n = 86), H395F (n = 113) and F391H/H395F (n = 83). **d** Store-operated Ca²⁺ entry (SOCE) signals measured by ratiometric fura-2 Ca²⁺ imaging in HEK-S1S2dKO-O1CFP cells transiently expressing STIM1-YFP WT (n = 16), or F391H (n = 52), H395F (n = 29) or F391H/H395F (n = 38) mutants. **e** Summary statistics for the average peak of SOCE entry in (**d**). STIM1-YFP-WT (n = 168), F391H (n = 207), H395F (n = 180) and F391H/H395F (n = 103). **f–j** Modification of the outer Phe-His pair. **f** Imaging of the ER–PM interface in HEK-S1S2dKO-O1CFP cells transiently expressing STIM1-YFP WT, or the F394H, H398F and F394H/H398F mutants thereof, 5 min after 2.5 μM ionomycin. Scale bar = 10 μm. **g** Time course of E-FRET interactions between Orai1-CFP and full-length STIM1-YFP-WT (n = 35), or F394H (n = 25), H398F (n = 17) or F394H/H398F (n = 25) mutants, before and after 2.5 μM ionomycin. **h** Summary data for E-FRET analyses in (**g**), as in (**c**) above. STIM1-YFP WT (n = 84), F394H (n = 99), H398F (n = 84) and F394H/H398F (n = 100). **i** SOCE signals measured by fura-2 imaging in HEK-S1S2dKO-O1CFP cells transiently expressing STIM1-YFP-WT (n = 14), or F394H (n = 12), H398F (n = 22) or F394H/H398F (n = 22) mutants thereof. **j** Summary statistics for the average peak of SOCE in (**i**). STIM1-YFP WT (n = 233), F394H (n = 96), H398F (n = 126) and F394H/H398F (n = 260). All data are presented as mean ± SEM of at least 3 independent experiments.

prevented STIM-Orai binding and hence also prevented channel activation, as expected. Indeed, an important role of F394 in Orai1 binding was reported earlier[33,37]. Unexpected, however, was that the combined F394H/H398F mutations reversed the action of the F394H mutation and restored full STIM-Orai1 binding. Thus, the reversed His-Phe pair appears able to take on the same configuration as the wildtype Phe-His pair to support binding. Moreover, even the Phe-Phe pair (in the H398F point mutation) permits normal STIM-Orai1 binding. But, despite the successful STIM-Orai1 binding supported by the His-Phe or Phe-Phe pairs, neither of these latter pairs permit the configuration necessary for Orai channel gating. This profound "uncoupling" of Orai1 binding from Orai1 activation indicates that the Phe-His pair within the short Sα2 helix holds the SOAR apex in a configuration that is

necessary for transducing Orai1 binding into channel activation. Clearly, the His residue at position 398 is critically important for this transduction event.

## The STIM1 His-398 residue has an indispensable role in functional coupling to Orai1

We sought to independently and more directly verify this apparently pivotal role of the STIM1-H398 residue in Orai1 coupling. For this, we utilized a novel optogenetically-engineered construct comprising the entire cytoplasmic STIM1 C-terminus (STIM1ct; residues 233–685), with the light-activated *Arabidopsis* cryptochrome 2 (CRY2) protein N-terminally linked, and C-terminally tagged with m-Cherry[27]. This construct undergoes light-activated coupling with Orai1,

independently of store-depletion and without the need for ER-PM junction formation. Upon 470 nm light-activation, the N-termini of this cytoplasmically-expressed STIM1ct cross-link, emulating but obviating the need for store-depletion, which in native dimeric STIM1 causes the ER luminal $Ca^{2+}$-sensing domains to associate to activate STIM1[19,23,26,27]. We transiently co-expressed this mCh-CRY2-STIM1ct construct, or the H398F mutant thereof, in HeLa cells, together with YFP-Orai1, and the

$Ca^{2+}$ probe GCaMP6s. Upon light activation, both the WT and H398F versions of the STIMct construct relocated to the PM, with a concomitant cytoplasmic loss (Fig. 2a, b). In both cases, the m-Cherry-tagged STIM1ct constructs were highly co-localized with YFP-Orai1 at the PM (Fig. 2c, d) indicating both bind strongly to Orai1. $Ca^{2+}$ levels measured with GCaMP6s (Fig. 2e, f) revealed that the WT-construct activated a large cytoplasmic $Ca^{2+}$ response after light activation,

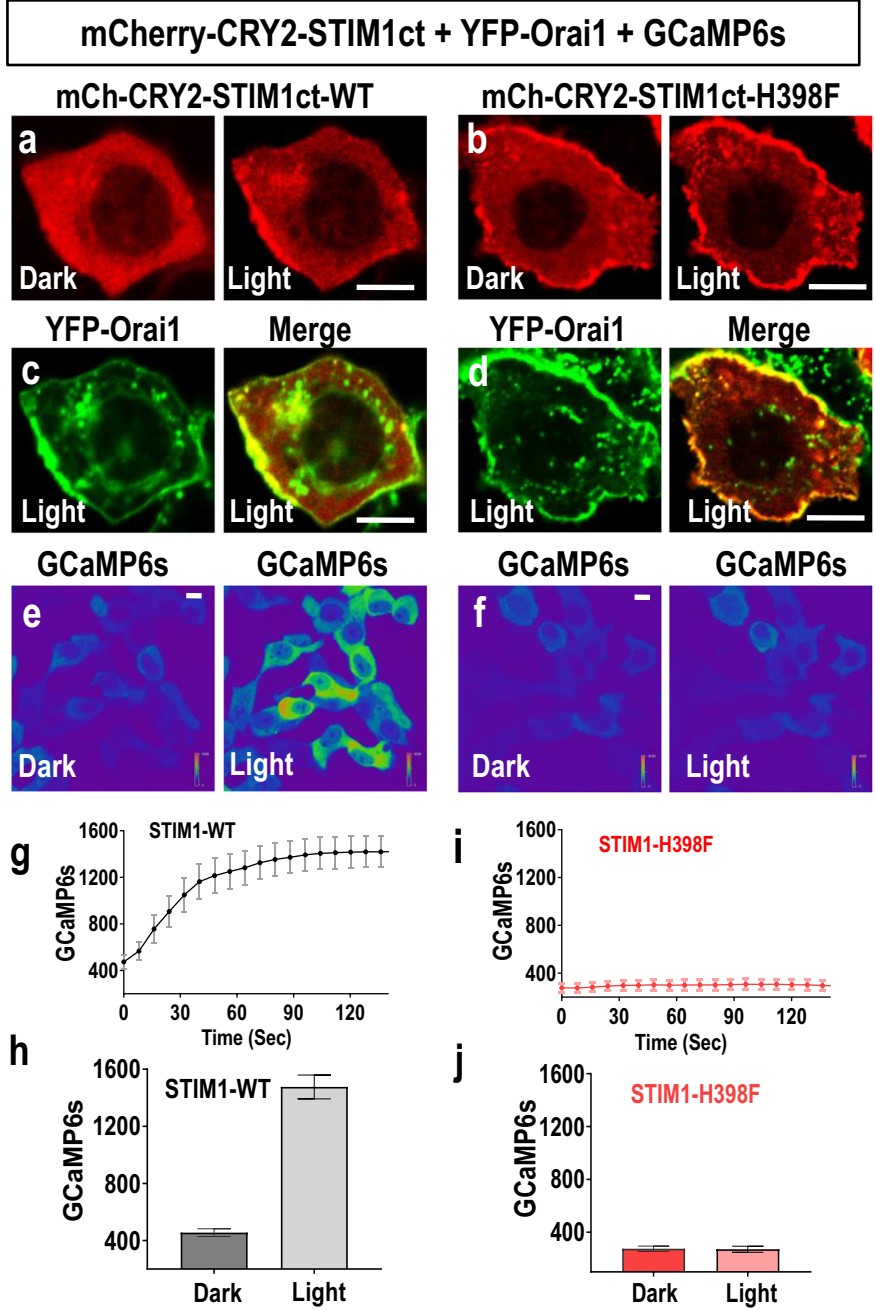

**Fig. 2 | Verifying the pivotal role of STIM1-H398 using an optogenetically-engineered STIM1 cytoplasmic C-terminal construct. a–d** Confocal images showing blue (488 nm) light-triggered co-localization of the light activated mCh-CRY2-STIM1ct construct (mCherry-tagged STIM1ct, residues 233–685, fused to CRY2) co-expressed with YFP-Orai1 and the $Ca^{2+}$ probe, GCaMP6s, in single Hela cells. **a** mCh-CRY2-STIM1ct-WT construct, or **b** mCh-CRY2-STIM1ctH398F mutant construct localization before (dark) and after (light) 488 nm illumination; **c, d** images of the same single cells in (**a**) and (**b**), respectively, showing co-expressed YFP-Orai1 localization after light activation (left), or merged images of YFP-Orai1 and mCh-CRY2-STIM1ct-WT (right). Images of multiple cells treated as in

(**a**–**d**), showing confocal light-induced GCaMP6s images of cytosolic $Ca^{2+}$ in HeLa cells co-transfected with GCaMP6s and YFP-Orai1, and either the CRY2-STIM1ct WT (**e**) or the CRY2-STIM1ct-H398F mutant (**f**). Cells were exposed to dark and blue (488 nm) light illumination for 2 min. **g** Time course of GCaMP6s $Ca^{2+}$ signals in response to photostimulation for mCh-CRY2-STIM1ct-WT (n = 23), representative graph from 3 independent repeats. **h** Summary statistics for the average peak of $Ca^{2+}$ signals shown in (**g**), WT (n = 72). **i** Time course of $Ca^{2+}$ signals after photo-stimulation for the mCh-CRY2-STIM1ct-H398F mutant (n = 30), representative graph from 3 independent repeats. **j** Summary statistics for the average peak of $Ca^{2+}$ signals shown in (**i**), H398F (n = 85).

whereas there was no detectable response with the H398F mutant (Fig. 2e–j). Thus, the crucial role of the H398 residue in the functional, as opposed to physical coupling between STIM1 and Orai1, is entirely replicated in a system independent of artificial store-depletion or the formation of ER-PM junctions.

The surprisingly critical role of the STIM1 H398 residue in gating the Orai1 channel prompted further mutational analysis. Mutation of H398 to the strongly positive residues, Lys or Arg, caused STIM1 to retain full Orai1-binding measured by FRET, but no detectable activation of Ca²⁺ entry (Fig. 3a, b) in HEK-S1S2dKO-O1CFP cells. Even substitution of H398 with the inert small amino acids, Ala or Gly, caused almost complete loss of Orai1 channel activation yet full retention of Orai1 interaction. The acidic residue, Asp, at this location resulted in similar loss of Ca²⁺ entry, but there was also some loss of Orai1 interaction. Aromatic substitution with Trp at 398 did support significant, albeit small Ca²⁺ entry, and FRET with Orai1 was reduced by approximately 30%. Likely, the indole ring in Trp can partially substitute for the imidazole in His. But, replacement of His-398 with the aromatic Phe residue, or reversal of the Phe-His pair (Fig. 1g–j) could not restore any Ca²⁺ entry, while binding was largely unchanged. Overall, the results substantiate the critical role of His at position 398 in *functional* coupling. H398 does not mediate the simple attachment of Orai1 per se, instead, H398 transduces the binding signal into functional coupling with Orai1. Thus, within the critical outer Phe-His pair, the F394 residue is essential in the physical binding of Orai1, and the H398 residue functions in tandem to impart conformational coupling on Orai1 to trigger channel gating.

### The STIM1 H398 residue mediates conformational switching in the Orai1 channel

Based on the critical role of the STIM1-H398 residue in functional coupling with Orai1, we sought to assess whether there was any concomitant conformational change in the Orai1 channel that might correlate with mutation of the STIM1-H398 residue. For this, we undertook FRET analysis using two C-terminally-tagged derivatives of Orai1, one with YFP and the other with CFP, co-expressed together with mCherry-STIM1 (either WT or H398 mutants) in HEK-S1S2dKO cells. Orai1 channels exist as hexamers, comprising three pairs of Orai1 proteins with asymmetric interactions between their adjacent outward-facing C-terminal M4x peptides[49]. These peptides form the initial, predominant STIM1 binding site[39]. The co-expression of both CFP- and YFP-tagged Orai1 derivatives can yield FRET measurements that reflect conformational alteration of the Orai1 channel C-termini as a result of STIM binding[38,50], depicted diagrammatically in Fig. 3c. The results of such an experiment (Fig. 3d, e) revealed that upon store-depletion with ionomycin, the activated wildtype STIM1 protein causes a rapid and substantial decrease in Orai-CFP/Orai-YFP FRET, likely as a result of a reconfiguration and probable unfolding of the Orai1 C-termini that represents the initial STIM1-induced conformational change in Orai1 leading to channel gating[7]. The replacement of mCherry-STIM1-WT with equally expressed mCherry-STIM1-H398F resulted in a greatly reduced rate and extent of Orai-CFP/Orai-YFP E-FRET interactions (Fig. 3d, e). Consistent results were obtained with expression of either the mCh-STIM1-H398K or mCherry-STIM1-H398R constructs which each have the same high Orai1 binding as wildtype STIM1 (Fig. 3a), but induce no Ca²⁺ channel activation (Fig. 3b; Supplementary Fig. 3). In a control experiment, we observed that mCherry-STIM1 carrying the F394H mutation (almost devoid of Orai1 binding; Fig. 1g), caused little change in Orai1-CFP/Orai-YFP FRET (Fig. 3d, e). In summary, these results provide evidence that the single STIM1-H398 residue plays a key role in transducing a conformational change in the Orai1 C-terminus, correlating with the ability of STIM1 to induce Orai1 channel gating. This conformational change is a consequence of the strong STIM-Orai-binding that occurs independently of mutation of the H398 residue. Thus, H398 mediates functional but not physical coupling to Orai1.

### The outer Phe-His residues in STIM1 must be paired to trigger Orai1 channel gating

To gain a better understanding of the functional role of the outer Phe-His pair within the Sα2 helix of the exposed SOAR apex of STIM1, we examined a series of Sα2 deletions and substitutions made in full-length STIM1 (Fig. 4a, b). Initially we deleted each of the outer Phe-His pair residues (Fig. 4c–e). Deletion of F394 caused complete loss of SOCE and

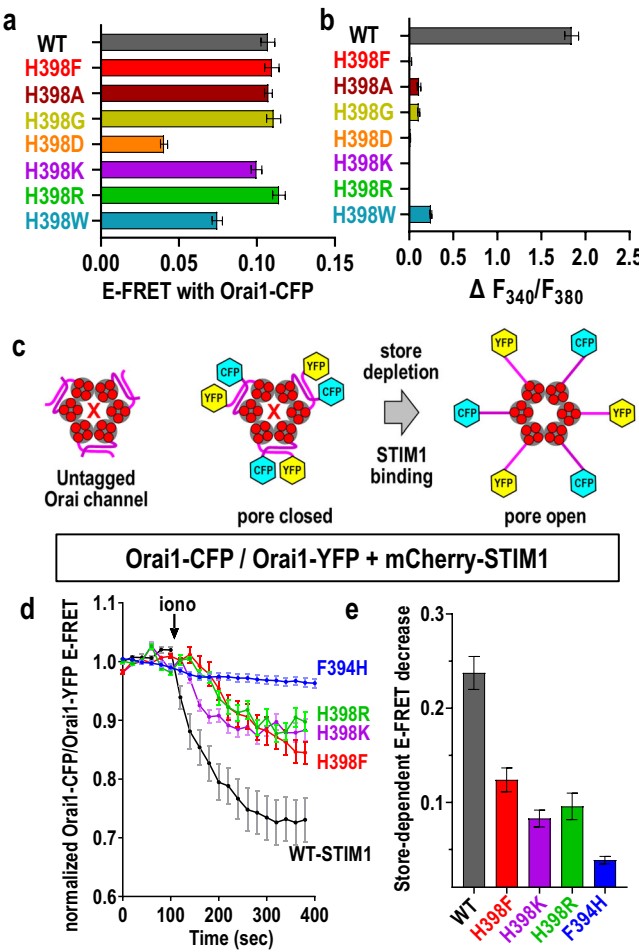

**Fig. 3 | The critical role of STIM1-H398 in Orai1 channel coupling, and its induction of a rapid configurational change in the Orai1 C-terminus. a** Summary statistics for peak E-FRET between Orai1-CFP and STIM1-YFP-WT (n = 132), or H398F (n = 86), H398A (n = 102), H398G (n = 65), H398D (n = 86), H398K (n = 121), H398R (n = 63) or H398W (n = 100) mutants thereof, in HEK-S1S2dKO-O1CFP cells after 5-min of store-depletion with 2.5 μM ionomycin. **b** Summary statistics for the average peak of SOCE in HEK-S1S2dKO-O1CFP cells transiently expressing STIM1-YFP WT (n = 106), or H398F (n = 78), H398A (n = 66), H398G (n = 84), H398D (n = 110), H398K (n = 80), H398R (n = 90), or H398W (n = 96) mutants thereof. **c** Model for YFP-Orai1-CFP-Orai1 FRET changes upon store depletion. The Orai1 hexameric channel is shown in its resting state with C-terminal transmembrane helix 4 extensions (M4x) in close proximity within Orai1 dimer pairs (left). Using cells co-transfected with both C-terminally tagged Orai1-CFP and Orai1-YFP, Orai1 hexamers contain mixed YFP/CFP tags (middle). Following store-depletion, dissociation of the M4x helices due to STIM1 binding leads to decreased Orai1-CFP Orai1-YFP FRET. **d** The E-FRET change measured in HEK-S1S2 dKO cells expressing mixed C-terminally-labeled Orai1-WT-CFP and Orai1-WT-YFP together with mCherry-STIM1 WT (n = 12) or H398F (n = 20), H398R (n = 11), H398K (n = 20), or F394H (n = 22) mutants thereof. E-FRET was observed in response to store-depletion with 2.5 μM ionomycin. E-FRET values were normalized to $t_0$. **e** Summary of YFP-CFP E-FRET changes induced by STIM1 WT and the H398 mutants used in (**d**), mCherry-STIM1 WT (n = 67) or H398F (n = 68), H398R (n = 35), H398K (n = 49), or F394H (n = 95). All data are presented as mean ± SEM of at least 3 independent experiments. Scale bars shown are 10 μm.

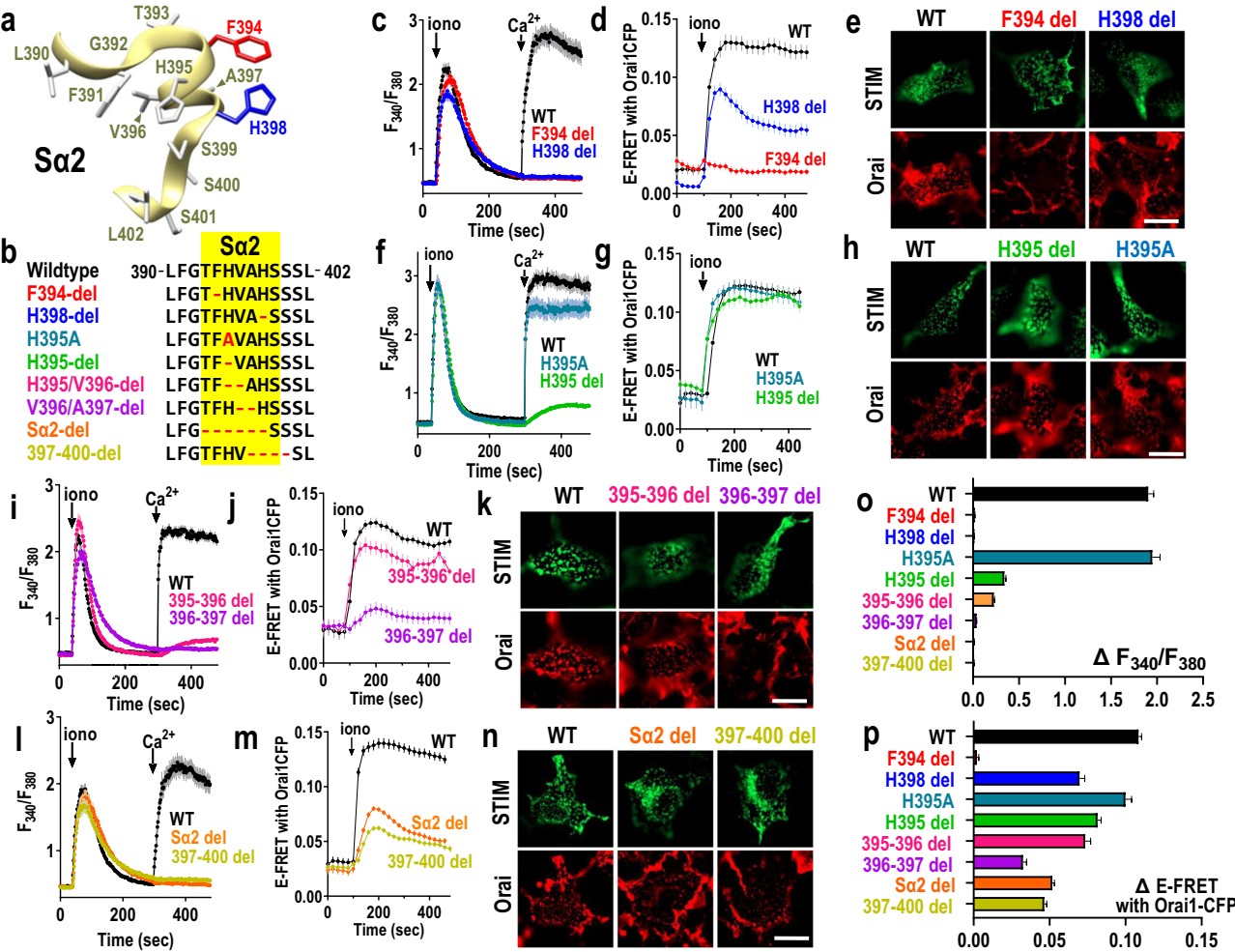

**Fig. 4 | Defining the essential role of residues in the SOAR Sα2 helix of STIM1 in coupling to the Orai1 channel. a** Schematic representation of the Sα2 helix region within SOAR. **b** sequence alignments of the Sα2 deletion or mutation constructs in full-length STIM1 expressed in HEK-S1S2dKO-O1CFP cells used in (**c**–**p**). **c** Cytosolic Ca²⁺ signals measured by fura-2 ratiometric Ca²⁺ imaging in HEK-S1S2dKO-O1CFP cells transiently expressing STIM1-YFP WT (n = 41), or the F394 (n = 35) or H398 (n = 39) deletion constructs, showing Ca²⁺ add-back after store-depletion with 2.5 μM ionomycin. **d** Time course of E-FRET interactions between Orai1-CFP and expressed STIM1-YFP WT (n = 38), or F394 (n = 33) or H398 (n = 34) deletions, before and after store-depletion with 2.5 μM ionomycin. **e** High-resolution imaging of the ER-PM interface in HEK-S1S2dKO-O1CFP cells transiently expressing STIM1-YFP WT, or the F394 or H398 deletions, measured 5 min after 2.5 μM ionomycin treatment. **f**–**h** Cytosolic Ca²⁺ signals, E-FRET, and co-localization measurements undertaken HEK-S1S2dKO-O1CFP cells as in (**c**–**e**), but instead using STIM1-YFP-WT (Ca²⁺ signals: n = 30; E-FRET: n = 46), or the STIM1-YFP-H395A (Ca²⁺ signals: n = 24; E-FRET: n = 27) point mutation, or the STIM1-YFP-H395 deletion (Ca²⁺ signals: n = 49; E-FRET: n = 52) mutant. **i**–**k** Cytosolic Ca²⁺ signals, E-FRET, and co-localization

measurements undertaken HEK-S1S2dKO-O1CFP cells as in (**c**–**e**), but instead using STIM1-YFP-WT (Ca²⁺ signals: n = 24; E-FRET: n = 42), or the STIM1-YFP-395-396 deletion mutant (Ca²⁺ signals: n = 40; E-FRET: n = 21), or the STIM1-YFP-396-397 deletion mutant (Ca²⁺ signals: n = 49; E-FRET: n = 26). **l**-**n** Cytosolic Ca²⁺ signals, E-FRET, and co-localization measurements undertaken HEK-S1S2dKO-O1CFP cells as in (**c**–**e**), but instead using STIM1-YFP-WT (Ca²⁺ signals: n = 29; E-FRET: n = 46), or the STIM1-YFP-Sα2 deletion (residues 393–398 deleted) (Ca²⁺ signals: n = 25; E-FRET: n = 39), or the STIM1-YFP-397–400 deletion (Ca²⁺ signals: n = 31; E-FRET: n = 49). **o** Summary statistics for the average peaks of store-operated Ca²⁺ entry in (**c, f, i, l**), STIM1-YFP WT (n = 232), F394 deletion (n = 108), H398 deletion (n = 97), H395A (n = 72), H395 deletion (n = 98), 395-396 deletion (n = 80), 396-397 deletion (n = 123), Sα2 deletion (n = 84), 397-400 deletion (n = 98). **p** Summary data for E-FRET analyses in (**d, g, j, m**). STIM1-YFP WT (n = 579), F394 deletion (n = 106), H398 deletion (n = 100), H395A (n = 91), H395 deletion (n = 207), 395–396 deletion (n = 87), 396–397 deletion (n = 107), Sα2 deletion (n = 212), 397–400 deletion (n = 217). All data are presented as mean ± SEM of data derived from at least 3 independent experiments. Scale bars shown are all 10μm.

FRET interaction with Orai1, despite normal full punctal localization within ER-PM junctions in response to store-depletion. Deletion of H398 resulted in strong FRET interaction and colocalization with Orai1, but the complete absence of any store-induced Ca²⁺ entry. These results establish further the indispensable role of F394 for binding to Orai1, however, the combination of F394 and H398 is required for transducing binding into the gating of Orai1. We next asked whether the 3-residue separation between F394 and H398, was also important. Thus, if F394 and H398 were operating as an aligned pair separated by a single helical turn (see Fig. 4a), then decreasing their intervening distance to less than one helical turn could prevent their interaction. Deletion of H395 to shorten the intervening distance by one residue, resulted in no loss of

STIM1-Orai1 binding or co-localization, but an almost complete loss of SOCE (Fig. 4f–h). In stark contrast, substituting H395 with Ala (H395A) did not affect STIM1-Orai1 association or colocalization, and allowed almost full SOCE (Fig. 4f–h). Elimination of the two intervening residues, H395 and V396, had a similar effect as deletion of just H395 (Fig. 4i–k). Elimination of the V396 and A397 intervening residues had quite a large inhibitory effect on STIM1-Orai1 binding and colocalization in addition to blocking any SOCE. As might be expected, deletion of the entire Sα2 helix (393-399) resulted in complete loss of SOCE activation, but interestingly, there was still some binding and co-localization of Orai1 (Fig. 4l–p). Similar results were seen with the shorter 397-400 deletion of the distal portion of Sα2.

The results demonstrate the remarkable molecular precision of the outer Phe-His pair in the apical Sα2 helix in transducing STIM1 binding into Orai channel activation. The Sα2 helix is adjacent to a short Sα3 helical turn (400-404; see Supplementary Fig. 1b) which includes the L402 residue. A recent study reported that the STIM-L402D mutation blocks Ca²⁺ entry without effects on Orai1 binding, and that the L402 residue mediates functional coupling to Orai1 as we describe here for H398[51]. However, FRET assessment to directly determine the interaction of the STIM1-L402D mutation with Orai1 was not undertaken. Our own studies with this mutation in HEK-S1S2dKO cells devoid of endogenous STIM proteins, revealed that in addition to eliminating Orai1 channel activation, the L402D mutation also substantially inhibited STIM1-Orai1 FRET (Supplementary Fig. 4). Thus, while the L402 residue in Sα3 clearly contributes to binding of Orai1, it does not mediate the essential functional coupling role seen here for H398.

## Positive charge at STIM1-394 triggers and negative charge blocks STIM1 unclamping

The above results reveal the remarkably precise requirement for the apical Phe-His pair in STIM1 for coupling with Orai1 to mediate channel gating. However, before this apical extremity of STIM1 can interact with Orai1 in the PM, the entire STIM1 cytoplasmic domain must undergo a substantial unfolding and elongation from its resting state, an activation process that is triggered by sensing lowered ER luminal Ca²⁺ levels[2,7,17]. Unexpectedly, we found that the same SOAR apical locus, which in the activated unfolded STIM1 state interacts with Orai at the PM, is held in the resting folded STIM1 state at a site that is deep within the folded STIM1 protein and close to the ER membrane. This important information arose from the surprising finding that mutation of the F394 residue to a positively-charged amino acid, causes a massive and complete unfolding of STIM1 such that its appearance is completely punctal and almost entirely within ER-PM junctions. Thus, as shown in Fig. 5a, upper, wildtype STIM1-YFP expressed in HEK-S1S2dKO cells exists within ER cisternae distributed throughout the cell. 5 min following store-depletion with ionomycin, STIM1 becomes almost completely punctal in appearance as it translocates and accumulates within ER-PM junctions (Fig. 5a, lower). In contrast, STIM1-YFP-F394K expressed under identical conditions, was almost entirely punctal in appearance without any store-depletion (Fig. 5b, upper). Analysis of punctal size revealed the puncta were slightly smaller compared to STIM1-WT, but the total punctal area in cells was similar (Supplementary Fig. 10). The absence of endogenous STIM protein expression, assured that this constitutive punctal distribution of STIM1-F394K was due to the mutation itself. Indeed, after ionomycin-induced store-depletion, the distribution of STIM1-F394K, remained little changed and still fully localized in puncta (Fig. 5b, lower). Almost the same constitutive punctalization was observed with STIM1-YFP-F394R (Fig. 5c). Thus, the positive charge at residue 394 induced complete constitutive unfolding and relocation of STIM1 into ER-PM junctions. In contrast, similar expression of STIM1-YFP-F394D (Fig. 5d, upper) or STIM1-YFP-F394E (Fig. 5e, upper), revealed no such constitutive puncta formation and the mutants were distributed throughout the ER under resting conditions. Importantly, with either of these two F394 acidic residue substitutions, no altered distribution of STIM1 into puncta was observed following store-depletion (Fig. 5d, e, lower). Thus, the unfolding and relocation of STIM1 into junctions is completely prevented by a negative charge at residue 394.

Despite the strong constitutive punctal reorganization induced by the substitution of F394 with positively charged residues in STIM1, there was no physical or functional coupling with Orai1. Thus, expression of STIM1-YFP-F394K in HEK cells stably expressing Orai1-CFP revealed an entirely distinct distribution pattern for the two proteins (Fig. 5f). While STIM1 was completely punctal, Orai1 was distributed within the PM with no localization in puncta (Fig. 5f upper).

Even after store-depletion, the distribution remained the same (Fig. 5f, lower). Examination of E-FRET revealed that neither STIM1-YFP-F394K nor STIM1-YFP-F394R had any measurable interaction with Orai1-CFP, compared to the very large E-FRET increase observe for wildtype YFP-STIM1 (Fig. 5g, i). Examining Ca²⁺ signaling in similarly transfected cells (Fig. 5h, j) revealed that neither the F394K nor F394R STIM1 mutants induced any coupling to overexpressed Orai1 to mediate Ca²⁺ entry. In a series of similar experiments with the F394D and F394E point mutations, it was clear that in addition to preventing any ability of these mutants to form puncta, there was also no physical coupling with Orai1 from E-FRET measurements (Fig. 5g, i), nor was there any observable activation of Orai1 to give Ca²⁺ signals (Fig. 5h, j). However, since the F394D and F394E mutations in full-length STIM1 could not be activated and could not translocate into ER-PM junctions to reach PM Orai1 channels, we needed to examine cytoplasmically-expressed YFP-SOAR constructs carrying each of the F394 mutants in HEK-S1S2dKO-OraiCFP cells, to assess any Orai1 co-localization, E-FRET with Orai1, or Orai1-mediated Ca²⁺ entry (Supplementary Fig. 5). As shown, YFP-SOAR-WT completely colocalized with Orai1-CFP, underwent strong FRET with Orai1-CFP, and induced large Orai1-mediated Ca²⁺ entry. In contrast, neither YFP-SOAR-F394D, YFP-SOAR-F394E, YFP-SOAR-F394K, nor YFP-SOAR-F394R showed any co-localization, E-FRET, or Ca²⁺ entry with Orai1-CFP. Thus, the effects of charged mutations on F394 on either activating (F394K, F394R) or blocking (F396D, F394E) STIM1 unfolding, were clearly not due to any altered interaction with Orai1 channels.

In summary, two powerful and very different phenotypes arise from positive vs. negative amino acid substitution at the key apical F394 residue. The positive charge causes an intriguing massive constitutive activation and junctional localization of STIM1, presumably reflecting unclamping of SOAR. Conversely, the negative charge completely prevents any activation or redistribution of STIM1 following store-depletion, and retards unclamping. Despite these effects, neither mutation allows any coupling with Orai1.

## STIM1 clamping or unclamping reflects a spatially precise charge on SOAR

Considering the remarkable specificity of the F394/H398 pair in functional coupling of STIM1 with Orai1 described above, we wished to determine more about the positional specificity of the powerful STIM1 unclamping and anti-unclamping effects induced by the F394K and F394E mutations, respectively. Therefore, we examined whether adding Lys or Glu residues at sites within the apical Sα2 helix, either adjacent or close to residue 394, would have similar effects. For this, we created point mutations at the T393, H395, and H398 residues in STIM1 (see Fig. 5k). The results were very clear. Thus, in contrast to the F394K mutation, substitution of either of the flanking residues with Lys (T393K or H395K) did not induce any punctal STIM1 under resting conditions, nor did they alter the formation of puncta in response to store-depletion (Fig. 5l). The T393K mutation slightly reduced and the H395K had no effect on store-induced Ca²⁺ entry (Supplementary Fig. 6). Similarly, substitution of the same two flanking residues with Glu (T393E or H395E) did not prevent the formation of STIM1 puncta induced by store-depletion seen with F394E, and in each case, the two mutants were still able to induce Ca²⁺ entry (Fig. 5l; Supplementary Fig. 6). Because of its close spatial proximity to F394 described above, we also examined substitution of H398 with either Lys or Glu on STIM1 localization (Fig. 5l). The STIM1-H398K mutation was entirely non-punctal in the resting state, and underwent normal redistribution into puncta after store-depletion. Similarly, the H398E mutation was entirely non-punctal and did not prevent store-induced puncta formation. Of course, adding either positive or negative residues to replace H398 gave rise to little effect on Orai1 binding but complete block of Ca²⁺ entry, as described above in Fig. 3a, b and Supplementary Fig. 3. Overall, we conclude that the apical F394 residue is critically

oriented within the STIM1-Sα2 helix within SOAR such that charge at this site drastically affects unclamping. The spatial specificity of these charge effects is extreme, and addition of charge even at the spatially close 398 position has no effect on unclamping.

### Positive charge at STIM1-394 replicates store-induced STIM1 unclamping

The powerful and specific effects of charge addition at the F394 site on STIM1 folding, prompted us to investigate more on the mechanistic basis of these unexpected findings. First, we asked how could the addition of positive charge (F394K) induce powerful translocation of STIM1 into ER-PM junctions when this same mutation completely blocks interaction with Orai1? Our question was whether the F394K mutation induced some unusual reconfiguration of STIM1 such that it

was accumulated nonspecifically in puncta, perhaps unrelated to ER-PM junctions. To answer this, we made a short C-terminal deletion in the STIM1-F394K construct to remove the polyK tail. This polyK region (residues 671–685) includes seven Lys residues and is known to allow STIM1, after store depletion-induced unfolding, to interact with acidic phospholipids in the PM and link it to the PM cytoplasmic surface, even with no Orai channels present[35,36]. Our results reveal that the constitutive punctal localization of STIM1-F394K was entirely prevented when the polyK sequence was deleted (Fig. 6a, b; Supplementary Fig. 10). This indicates that the F394K mutation is replicative of store depletion-induced activation of STIM1 allowing it to accumulate within ER-PM junctions where the polyK region tethers it to the PM. Without the ability to bind PM phospholipids, the STIM1-F394K mutant remains distributed within ER cisternae with no visible

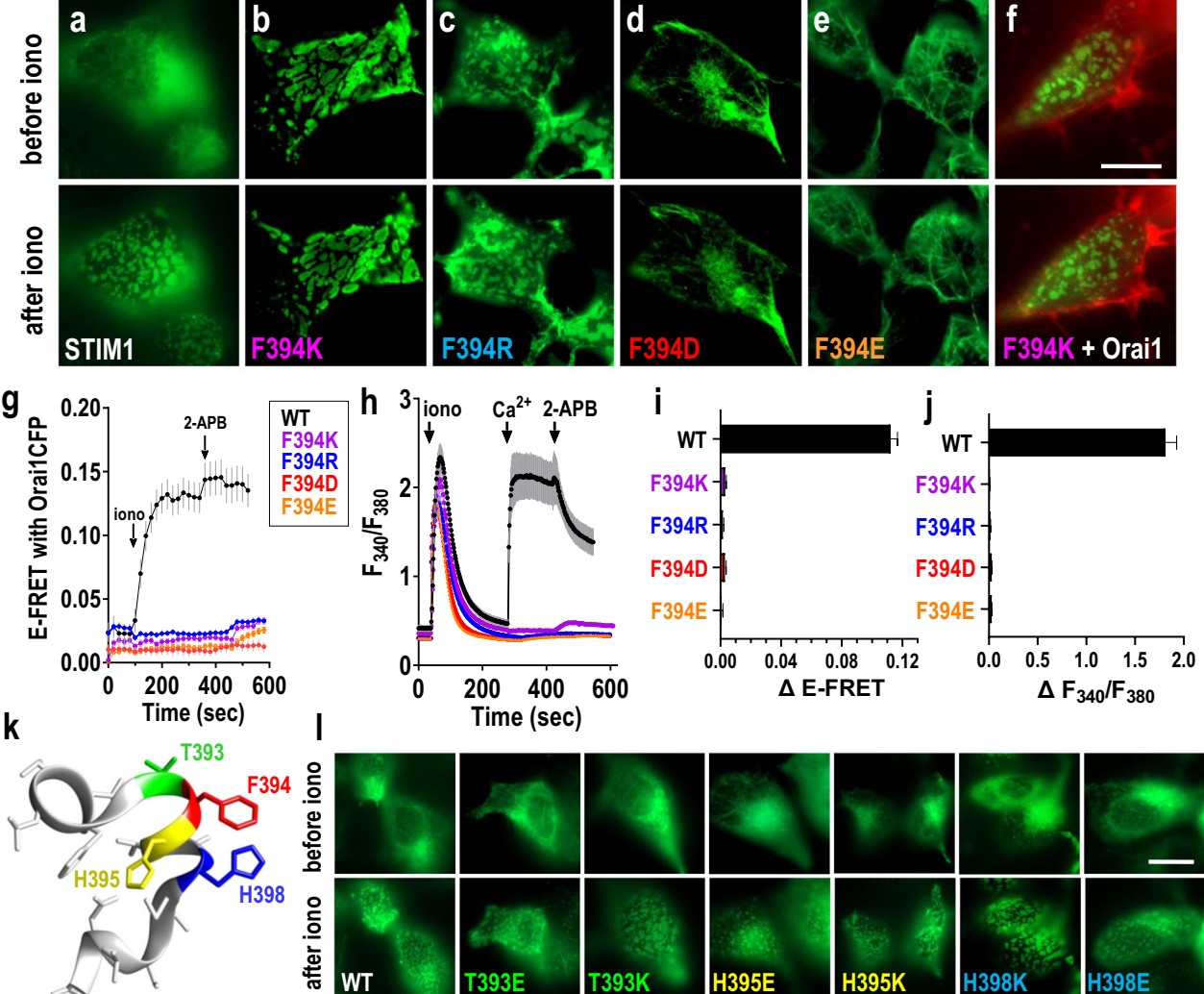

**Fig. 5 | Negative and positive charge addition to the F394 residue in the STIM1 Sα2 apical helix cause completely opposite effects on the clamped state of STIM1. a–f** High-resolution imaging of the ER–PM interface in HEK-S1S2dKO cells transiently expressing STIM1-YFP-WT (**a**), or the F394K (**b**), F394R (**c**), F394D (**d**), F394E (**e**) mutants of STIM1, in each case showing STIM1-distribution either before (top) or after (bottom) store-depletion with 2.5 μM ionomycin. **f** High-resolution imaging of the ER–PM interface in HEK-S1S2dKO-O1CFP cells transiently expressing STIM1-YFP F394K, to reveal lack of overlap of Orai1-CFP and STIM1-YFP-F394K, both before (top) and after (bottom) 2.5 μM ionomycin. **g** Time course of E-FRET interactions between Orai1-CFP and transiently expressed STIM1-YFP WT (n = 21), or the F394K (n = 16), F394R (n = 43), F394D (n = 38) or F394E (n = 46) mutations of STIM1, before and after store depletion with 2.5 μM ionomycin. **h** Cytosolic Ca²⁺ signals measured by fura-2 Ca²⁺ imaging in HEK-S1S2dKO-O1CFP cells transiently

expressing STIM1-YFP-WT (n = 15), or the F394K (n = 29), F394R (n = 19), F394D (n = 23) or F394E (n = 56) mutants of STIM1. **i** Summary data for change in E-FRET analyses from (**g**), STIM1-YFP-WT (n = 134), or the F394K (n = 122), F394R (n = 142), F394D (n = 131), F394E (n = 162). **j** Summary statistics for the average peak of SOCE in (**h**), STIM1-YFP-WT (n = 68), or the F394K (n = 63), F394R (n = 151), F394D (n = 110), F394E (n = 170). **k** Schematic representation of the Sα2 helix within SOAR to highlight the T393, F394, H395 and H398 residues. **l** High-resolution imaging of the ER-PM interface in HEK-S1S2dKO-O1CFP cells transiently expressing STIM1-YFP-WT, or the T393E, T393K, H395E, H395K, H398K or H398E STIM1 mutations in the STIM1-YFp construct, in each case either before (top) or after (bottom) store-depletion with 2.5 μM ionomycin. All data presented are mean ± SEM of at least 3 independent experiments. Scale bars shown are all 10 μm.

punctalization, even after store-depletion. This is a significant finding, revealing that the F394K mutation does not directly induce aggregation of STIM which could then lead to accumulation within junctions. Thus, even though STIM1 is in its unfolded, activated state, its appearance in puncta is completely dependent on its ability to tether to the PM through the C-terminal polyK sequence. Interestingly, this also reveals that the C-terminal YFP tag, placed immediately adjacent to the poly-K region, does not prevent association of the poly-K sequence with acidic phospholipid headgroups on the PM. A recent study revealed that the EB-1 binding site on STIM1 (the "TRIP" domain, residues 642–645), stabilizes the STIM1 inactive folded state by enhancing its interaction with microtubules in the vicinity of the ER membrane[52]. However, elimination of the TRIP motif

did not reverse the inability of the STIM1-F394K-ΔpolyK to form puncta (Supplementary Fig. 8).

A further question was whether addition of charge at the F394 residue might indirectly alter the sensitivity of the Ca²⁺-sensing luminal domain and hence induce STIM1 activation. To approach this, we utilized the powerful STIM1 mutation, D76A, which reduces the luminal EF-hand affinity for Ca²⁺, and fully activates STIM1 without depletion of Ca²⁺ within stores[34]. We examined whether this mutation might overcome the action of F394D to block STIM1 activation. Indeed, this was not the case. The D76A point mutation within STIM1-YFP resulted in full constitutive activation of STIM1 with a completely punctal distribution that was not altered by store-depletion (Fig. 6c; Supplementary Fig. 10). In contrast, the double mutant (D76A, F394D) was

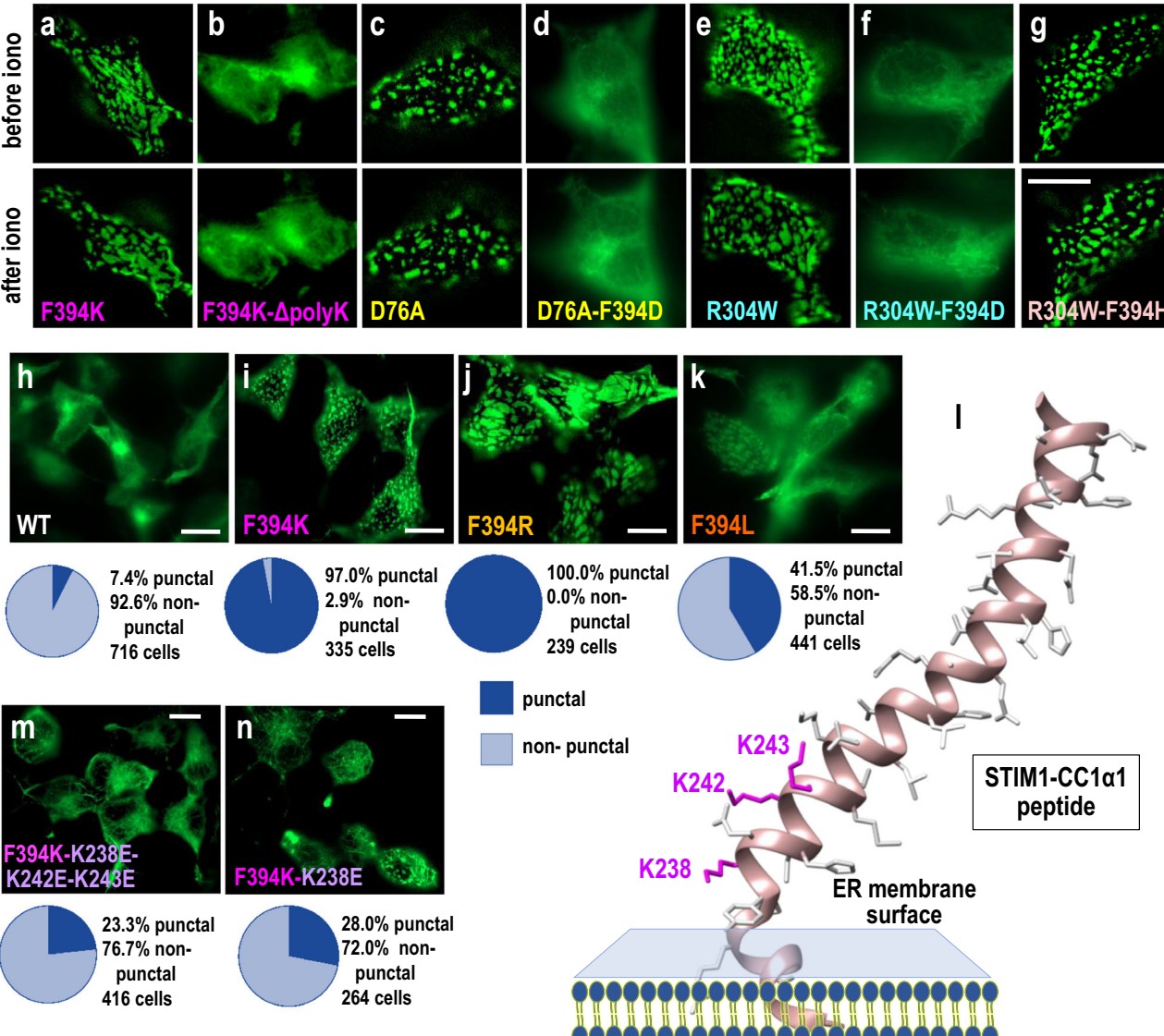

**Fig. 6 | Powerful changes induced by charge addition to the STIM1 Sα2 apical F394 residue, and the clamped position of F394 charge repulsion. a–g** High-resolution imaging of the ER-PM interface in HEK-S1S2dKO cells transiently expressing STIM1-YFP with the following mutations: **a** F394K, **b** F394K-ΔpolyK, **c** D76A, **d** D76A-F394D, **e** R304W, **f** R304W-F394D, and **g** R304W-F394H. In each case images shown are before (top) or after (bottom) store-depletion with 2.5 μM ionomycin. High-resolution imaging of the ER-PM interface in HEK-S1S2dKO cells transiently expressing either STIM1-YFP-WT (**h**), STIM1-YFP-F394K (**i**), STIM1-YFP-F394R (**j**), or STIM1-YFP-F394L (**k**) point mutation constructs. Note, cells in (**h–k**) were not treated with ionomycin. Beneath each image are pie charts summarizing

the percentage of punctal and non-punctal cells. **l** Schematic representation of the STIM1-CC1α1 helix including its insertion in the ER membrane (the STIM1 TM helix ends at 234). The three Lys residues (L238, L242, and L243) close to the ER, are highlighted (from PDB ID 6YEL). High-resolution imaging of the ER-PM interface in HEK-S1S2dKO cells transiently expressing either the STIM1-YFP-F394K-K238E-K242E-K243E quadruple mutation (**m**), or the F394K-K238E double mutation (**n**). Cells were not treated with ionomycin. Beneath are pie charts that summarize the percentage of punctal and non-puncta cells. The numbers of cells shown are total numbers derived from at least 3 independent transfections. Scale bars shown are all 10 μm.

entirely non-punctal and remained so even after store-depletion (Fig. 6d). Thus, the apical F394D mutation is able to completely prevent function of the highly effective D76A EF-hand mutation that normally induces full activation and translocation of STIM1 by short-circuiting its physiological activation mechanism. This action reflects the strength of the negative charge added to the F394 position in preventing unfolding of the STIM1 cytoplasmic domain.

### STIM1 unclamping by the human R304W GoF mutation, is blocked by F394D

The unclamping, unfolding and activation of STIM1 has major physiological significance and recent studies provide some mechanistic insight into this process[24,25]. In resting wildtype STIM1, the three CC1 helices fold such that the CC1α1 helix binds to the Sα4 helix of SOAR, clamping SOAR within the inactive STIM1 structure (see Supplementary Fig. 1). Store-depletion causes unfolding and extension of the CC1 helices allowing SOAR to flip out from the ER and engage PM Orai channels within ER-PM junctions[25,43]. This unfolding can also be activated by a naturally occurring gain-of-function mutation in human STIM1, R304W, which causes Stormorken syndrome characterized by multiple defects including platelet dysfunction, thrombocytopenia, anemia, tubular aggregate myopathy, and congenital miosis[53,54]. The mutation induces constitutive unfolding of STIM1 and an unchecked activation of Orai channels[24,25,28]. The R304 residue lies at a key position in CC1, and its mutation to Trp destabilizes the close packing of the CC1α2/CC1α3 helices, allowing the CC1α1 helix to unfold from the complex and release SOAR[25]. We examined how alteration of the F394 headgroup might affect the R304W-induced activation of STIM1. As shown in Fig. 6e and Supplementary Fig. 10, the STIM1-R304W mutation gave rise to a constitutively complete punctal distribution, little changed by store-depletion. Using the double STIM1-R304W-F394D, the R304W-induced punctalization is completely prevented by the F394D mutation (Fig. 6f). Indeed, even after store-depletion, there is no sign of STIM1 redistribution. This reversal of action of the combined effects of the GoF R304W mutation together with store-depletion, again reflects the strength by which the F394D mutation holds the clamped state of STIM1 and prevents any translocation into ER-PM junctions. Interestingly, mutation of the F394 headgroup to His had no suppressive effect on R304W-induced puncta formation (Fig. 6g; Supplementary Fig. 10). Thus, although both F394H and F394D both block the STIM1-Orai1 interaction, their respective effects on STIM1 unfolding are very different. Indeed, although the F394 headgroup plays a decisive role in both the internal unclamping of STIM1 and the binding of STIM1 to Orai1, the two processes are distinct. Unclamping is entirely blocked by negatively charged Asp and Glu residues, fully activated by positively charged Lys or Arg residues, and unaffected by His substitution at the 394 position. In contrast, the STIM1-Orai1 interaction is blocked by substitution with any of these residues at position 394.

### Unclamping of STIM2 may reflect an apical Leu instead of Phe

The degree to which unfolding and reorganization of STIM1 is induced by introduction of a positive charge at the SOAR apex was surprising. Expressed in HEK cells, wildtype STIM1-YFP can be seen as having a full punctal appearance in approximately 7% of cells (Fig. 6h) which may reflect ER Ca²⁺ leak in a small proportion of cells caused by STIM1 overexpression. In contrast, in cells expressing STIM1-YFP with either of the F394K or F394R point mutations, almost 100% of cells exhibited a punctal distribution (Fig. 6i, j). Interestingly, STIM2 has a highly conserved SOAR region, but the position equivalent to F394 in STIM1, is a Leu residue (L485) in STIM2. To examine this difference, we replaced the F394 residue in STIM1-YFP with Leu. This resulted in more than 40% of cells displaying a constitutively punctal STIM1 distribution (Fig. 6k). The non-punctal cells were still able to undergo full punctalization in response to store-depletion (Supplementary Fig. 7).

Previously, we reported that the same F394L point mutation introduced within SOAR from STIM1, resulted in a lowered efficacy of interaction with and activation of Orai[33]. Now, introduced within full-length STIM1, we reveal that this point mutation results in a considerably increased degree of spontaneous unfolding and reorganization of STIM1 into ER-PM junctions. Although we have not yet examined the activation and unfolding of full-length STIM2 protein, it has been well-documented in a number of studies that STIM2 has a substantially higher level of constitutive activation[55,56]. This has been generally attributed to a slightly lower Ca²⁺ affinity of the luminal EF-hand domain[55], however, the current results indicate that STIM2 may have a greater intrinsic propensity to transition into an unfolded, active conformation due to the Leu at the key position equivalent to F394 in STIM1.

### Determining the site of positive charge-induced unclamping of STIM1

Lastly, our studies provide an explanation for the extraordinary unfolding of STIM1 induced by a positively charged headgroup added at the key, apical 394 position within SOAR. More significantly, the results reveal the location of the SOAR apex within the folded structure of the resting state of STIM1. SOAR is thought to interact with the CC1α1 helix in the resting STIM1 molecule[18–25]. Initially, the helices were thought to interact in an antiparallel configuration[23], but recent evidence suggests a parallel interaction[25]. With this in mind, we noted that the CC1α1 helix includes three Lys residues (at positions 238, 242, and 243) close to its N-terminal end, and starting just 4 residues beyond the transmembrane helix (residues 214–234), close to the ER membrane surface (see Fig. 6l). Starting with the constitutively punctal STIM1-F394K-YFP construct, we initially reversed the charge on all three Lys residues, mutating them to Glu residues to give the STIM1-F394K-K238E-K242E-K243E-YFP construct. This gave rise to a substantial decrease in the level of constitutively punctal STIM1, from 97% for STIM1-F394K, to 23% for the STIM1-F394K-K238E-K242E-K243E-YFP construct (Fig. 6m). We continued our assessment by narrowing the reversal of charge on the three CC1α1 Lys residues to just the K238 position. Thus, using the STIM1-F394K-K238E construct we still observed a substantial decrease in the proportion of spontaneously punctal cells, in this case to just 28% of total cells (Fig. 6n). The K238E mutation alone did not affect basal STIM1 puncta formation, nor did it alter store-induced puncta formation or the ability of STIM1 to FRET with Orai1 (Supplementary Fig. 9). The results reveal a large decrease in the spontaneous unfolding and translocation of the STIM1-F394K mutation by simply reversing charge on the K238E mutation. This indicates that the F394 and K238 residues are positionally close, and predicts that in its resting state, the active apical headgroup of SOAR is held at a position deep within the folded STIM1 protein. Thus, as discussed below, we conclude the unfolding of STIM1 effected by Lys substitution at the apical F394 position is caused by charge repulsion at the K238 position, and therefore indicates the position of the SOAR apex in the resting STIM1 protein is a site very close to the ER membrane surface (see Supplementary Fig. 11).

## Discussion

Our results reveal information on the molecular precision of the STIM1 active site that triggers Orai1 channel gating. Moreover, they show how deeply concealed this active site is within the folded resting state of STIM1, and indicate its location in the resting STIM1 protein is at a site very close to the ER membrane. Hence, during activation of STIM1, the apical helix of the SOAR moiety of STIM1 undergoes a remarkable journey of about 10–15 nm, from close to the ER membrane surface in the folded state, to making direct contact with the Orai channel at the PM in the activated state (see Fig. 7).

The activation of STIM1 by release of the intramolecular "clamp" that exists in the folded STIM1 resting state, has been described in a

 

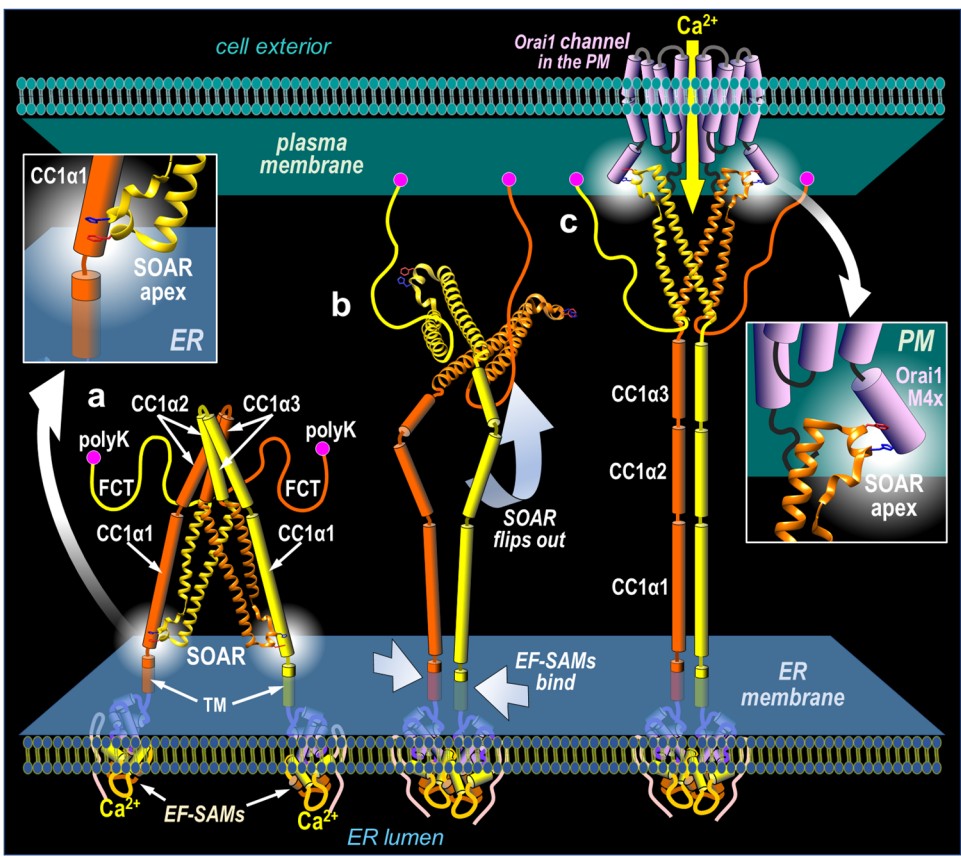

**Fig. 7 | Model for the STIM1 activation and Orai1 coupling cycle.** The scheme depicts the events following ER luminal Ca²⁺ depletion. **a** Resting state of the STIM1 dimer in Ca²⁺ replete ER showing the luminal Ca²⁺-sensing EF-SAM domains and transmembrane TM regions are separated, and the cytoplasmic helixes CC1α1, CC1α2, and CC1α3, are in a tightly folded configuration which clamps the STIM-Orai activating region (SOAR) and keeps it occluded from the cytosol. The flexible C-terminal (FCT) domains ending in polyK regions, are shown with no interactions. **b** In response to luminal ER Ca²⁺ depletion, the EF-SAM and TM regions bind together, triggering SOAR to be "squeezed" and released from the unfolding CC1α helices. SOAR is shown flipping out to the rear of the protein. Upon diffusion of STIM1 into ER-PM junctions, the FCTs allow the PK regions to attach to lipids on the PM surface, trapping STIM1 in junctions. **c** The CC1α helices zip together, and the fully extended STIM1 protein allows the SOAR domain to tether and gate PM Orai1 channels to mediate Ca²⁺ entry. The model highlights the importance and passage of the SOAR apex containing the outer Phe-His pair which, in the resting STIM1 protein is bound to the lower CC1α1 helix very close to the ER membrane (left inset), but in the activated STIM1 protein is bound to the C-terminal M4x helix immediately adjacent to the PM. The His-398 residue is key to transducing STIM1 binding into gating of the Orai1 channel.

number of recent reports[18–25]. This clamp is known to occlude the SOAR moiety within the folded cytoplasmic domain of the resting STIM1 protein and prevent its unintended exposure and coupling with Orai channels. Several different molecular arrangements have been described to account for this clamp. The more recent of these have defined an interaction between the CC1α1 helix and the Sα4 helix in SOAR as mediating the inhibitory clamp, involving the interactions of specific Leu residues in each of the two helices[19–25]. Most recently, elegant single molecule FRET studies[25], indicate that within the folded resting STIM1 dimer, CC1α1 and Sα4 helices interact together in a parallel arrangement, but with a remarkable inter-subunit "domain-swapping" configuration (see Supplementary Fig. 11). Hydrophobic interactions between Leu residues in CC1α1 (L248, L251, L258, L261) and Sα4 (L416, L423, L427) are believed to mediate these interactions. This unusual inter-subunit locking of helices may stabilize the folded inactive state, and perhaps promote a cooperative unfolding of STIM1 during store depletion-induced activation[43]. The single molecule FRET results of van Dorp et al. [25] also reveal that the SOAR dimer base is rigidly held within the STIM1 dimer. Interestingly, this study further revealed that the two SOAR apical regions have quite some degree of flexibility (see Supplementary Fig. 11), hence the apices can modestly deviate from the crystal-structure derived earlier from an isolated SOAR fragment[40]. As shown, the interacting Leu residues that likely clamp the CC1α1 and Sα4 helices are located toward the rigid SOAR base. Based on these interactions, our data indicating that the apical Sα2 F394K mutant residue is located close to the CC1α1 K238 residue, would not be predicted if the two helices were rigidly oriented (Supplementary Fig. 11). However, flexibility of the SOAR apices would allow for this interaction to occur, and we hypothesize that the apical regions containing the F394 residue, fold around the CC1α1 helices in order to stabilize SOAR close to the ER membrane surface in the resting state of STIM1. This is a most interesting scenario, since the same flexible apices likely fold around the C-terminal M4x peptide of the Orai1 channel and tether the activated SOAR unit close to the PM surface in order to gate the Orai1 channel[39,57]. Our recent studies with whole Orai1 channels and M4x peptides reveal that the Leu-273 and Leu-276 residues in Orai1 (and equivalent Leu residues in Orai2 and Orai3), thought previously to hold adjacent TM4-ext peptides in a binding pocket for SOAR, are in fact directly interacting with SOAR. Mutation of these residues on Orai channels mimics the F394 mutation of STIM1[39], however, whether STIM1 F394 interacts directly with these Orai channel M4x Leu residues, has yet to be determined. We had earlier suggested that two Orai1 His residues in close proximity to the M4x peptide, H264 in the adjoining "nexus" region, and H165 in the 2–3 loop, were important sites through which STIM1 could conformationally couple to trigger Orai1 gating[38]. It is possible that the STIM1 H398 residue, revealed here to be critical for functional coupling, might undergo aromatic pairing with such Orai1 His residues to

mediate channel activation. The model in Fig. 7 depicts the two SOAR units in the STIM1 dimer, interacting with adjacent Orai1 subunits within a single Orai1 channel. We revealed earlier[37] that each SOAR unit within the STIM1 dimer interacts independently with Orai1 subunits. Hence, the two SOAR units within a single STIM1 dimer are each able to bind to Orai1 subunits in separate Orai1 channels, causing cross-linking between Orai1 channels.

Our results reveal that the SOAR apical F394 headgroup is critical in the two essential functions of the STIM1 cytoplasmic domain – release of the SOAR clamp, and tethering of SOAR to activate Orai channels. Substitution of a charged residue at the 394 position drastically alters both these functions. Orai1 channel-coupling is completely eliminated with either positive or negative charge substitution at position 394. In sharp contrast, unclamping of SOAR is fully activated by a positive charge at position 394, whereas a negative charge completely blocks the unclamping. Our results suggest that a positively charged Lys residue added at the 394 position in Sα2 undergoes charge repulsion with the existing K238 residue in CC1α1, leading to a constitutive unclamping and relocation of STIM1 entirely into ER-PM junctions. In contrast, we observed STIM1-F394D is strongly locked in the clamped position and resists unfolding by either ER store-emptying, lowering luminal EF-hand $Ca^{2+}$ binding, or the powerful R304W GoF mutation. A simple explanation for this extraordinary divergence of effects on SOAR clamping in response to positive and negative charges added at 394, was that STIM1-F394D might form a salt-bridge with K238 giving rise to the powerful clamp lock. Hence, we examined whether Glu added at the 238 position would reverse the F394D-induced block of unclamping. However, this was not the case, and the STIM1-F394D-K238E mutant was still non-punctal and still did not become punctal upon store-depletion (Supplementary Fig. 12). This suggests that the powerful F394D mutation is enhancing the clamp though a mechanism not simply the converse of the F394K mutation. Possibly, the F394D mutation may more substantially modify the flexible SOAR apex and/or the nearby Sα4 helix in SOAR such that its association with CC1α1 is stronger. Interestingly, a recently published report used MD simulations to assess theoretical interactions between the SOAR and CC1α1 helices from STIM1, and noted that the F394D mutation strongly enhanced these helical interactions[58]. Curiously, however, another recently published study assessing FRET between cytoplasmically expressed SOAR and a truncated STIM1(1–342) construct exposing the CC1 helices on the ER surface, observed the F394D mutation completely blocked the interactions between these two constructs, indicating F394D would greatly promote unclamping[59]. We suspect that SOAR, in isolation from holo-STIM1, may acquire a quite different configuration in the cytoplasm as opposed to that it adopts deeply buried within the highly folded, resting STIM1 protein. And the same may be true for MD simulations of the peptides in isolation. This is another important reason why our studies used full-length STIM1 in which the SOAR units are being clamped or released in the natural molecular environment within the folded STIM1 protein.

Recent molecular simulation of the SOAR apical region suggested that the exposed F394 position itself might be interacting with lipid headgroups of the ER membrane[58]. Indeed, molecular simulations with the charged F394D and F394K mutations indicated that they might both enhance such membrane lipid interactions of SOAR. The models predicted F394K had the strongest interaction with charged ER membrane lipid surfaces, and concluded this mutation would cause SOAR to be tightly clamped and resist activation and unfolding. Our findings provide a stark contrast to this prediction. Thus, our analyses of full-length STIM1 functioning within living cells, reveal the F394K and F394R mutations cause exactly the opposite effect to that predicted, and powerfully unfold virtually 100% of expressed STIM1. Moreover, we reveal the F394D and F394E mutations have the converse effect to F394K and F394R, and tightly clamp STIM1 so it cannot be activated, despite F394D being predicted to have weaker

membrane interactions than F394K[58]. We are therefore cautious about applying molecular simulations to a dynamic protein such as STIM1, that is tightly folded with SOAR buried close to the ER membrane. Our co-mutational findings reveal that the complete unclamping of whole STIM1 by the F394K Sα2 helix mutation is largely prevented by the K238E mutation in the CC1α1 helix, indicating charge repulsion and probable close contact between the SOAR apex and the CC1α1 238 residue. Indeed, the latter residue is only one helical turn (4 residues) away from the 233 residue that defines the limit of the single transmembrane domain of STIM1. Thus, it seems that the SOAR apex may be folded within STIM1 very close to the ER membrane surface, but based on the unfolding induced by the F394K mutation, is not actually making contact with or holding SOAR to the membrane surface.

A recent report published while our studies were being completed[58] suggested that the STIM1 Sα2 helix was at least partly dispensable in the activation of Orai1 channels. Our studies used stable, CRISPR-derived HEK cells devoid of STIM1 and STIM2[38] and hence no interfering effects of endogenous STIM proteins. Using these, we reveal that gating of Orai1 channels is entirely prevented by, (a) deletion of the Sα2 helix, (b) deletion of just the H398 residue, (c) most mutations of H398, or (d) changes in the outer Phe-His pair distance. Thus, the Sα2 helix is key to transducing the STIM1-binding signal into gating of Orai1. Of course, modifications or even deletion of Sα2, have only partial effects on STIM1-Orai1 binding, thus we conclude STIM1-Orai1 binding involves a considerably larger segment of SOAR than just the Sα2 helix.

Our studies reveal the pivotal role of the spatially close Phe-394 and His-398 residues within the crucial SOAR apex of STIM1. This pair of residues are highly conserved in the SOAR apex of all mammalian STIM1 proteins, and even in the single STIM protein in *Drosophila* (see Supplementary Fig. 13). In resting STIM1, this region is clamped deeply within the folded CC1 helices, close to the ER membrane. The extreme effects of charge applied to the F394 position are remarkable. The complete induction of STIM1 unfolding with a positive charge or complete block of unfolding by a negative charge at the 394-residue, reflect the extraordinarily critical nature of the immediate environment of the SOAR apex within the clamped resting STIM1 protein. This folded environment for SOAR appears very different to that after STIM1 activation and exposure of SOAR within the cytoplasm. In this unclamped state, we reveal that the His-398 residue becomes critical for successful coupling to induce gating of the Orai1 channel. Yet, within the partially flexible SOAR apex[25], the spatially close Phe-394 residue has a dominant role in determining STIM1 binding to Orai1, and likely in the "presentation" of His-398 within the binding complex. We observe that mutations altering the Phe-His distance in STIM1 prevent Orai1 gating, whereas intervening residue substitution is not critical. Hence, we consider that the Phe-His residues interact within an aromatic pair to restrain the partially-flexible SOAR apex to optimize the conformation that will trigger Orai1 channel gating. Indeed, pairing of such aromatic headgroups adjacently aligned across a single helical turn with three intervening residues, is a common motif for preserving a precise helical stability and conformation in proteins[45,46]. The occurrence of two such Phe-His pairs in the SOAR apex is even more intriguing, especially since they also overlap. The distances and offset angles of these pairs are in keeping with structurally significant interactions within Phe-His pairs in many proteins[44–46] (Supplementary Fig. 14). In STIM2, while the critical His residue in the SOAR apex is conserved, a Leu residue (L485) replaces the equivalent position of F394 in STIM1. Thus, the outer Phe-His pair is disrupted in STIM2 and the local environment of the His residue is altered. This important difference likely explains the lower efficacy of STIM2-induced Orai1 activation compared to STIM1[16,33,60], and may contribute to the considerably greater degree of constitutive unfolding observed for STIM2[55,56,61] as revealed in Fig. 6k. Thus, key differences between STIM1 and STIM2 in the

stability, kinetics and efficacy of Orai1-binding result from this strategic subtype difference in the SOAR apex.

Lastly, our work throws light on the highly studied yet enigmatic action of the small diphenyl molecule, 2-APB, on STIM-Orai function. 2-APB affects both the unclamping of STIM proteins as well their coupling with Orai1 channels, and we suggest both effects are via the apical Phe-His residues. Earlier we showed 2-APB hugely promotes the constitutive (store-independent) activation of STIM2, with little effect on STIM1[47,62]. We also reported that 2-APB unclamps cytosolically expressed whole C-terminus of STIM2 (STIM2ct) much more effectively than STIM1ct, and facilitates binding to the Orai1 channel[47]. We observe here that an unpaired outer His residue, as present in STIM2, weakens clamping (Fig. 6k). The unpaired His in STIM2 may be the target through which 2-APB promotes complete STIM2 unclamping, whereas the strongly-clamped STIM1 with its paired outer Phe-His, is resistant to 2-APB-induced unclamping. With regard to STIM1-Orai1 coupling, we reveal here that 2-APB restores both the defective Orai1 binding and channel gating seen when either the inner or outer Phe-His pairs are converted to His-His pairs (Fig. 1a–i). We suggest that the diphenyl of 2-APB may form close Phe-His aromatic interactions with one or both His residues and hence mediate the successful coupling between STIM1 and Orai1 (see Supplementary Fig. 14). Interestingly, key His residues mediate 2-APB binding and its powerful activation of TRPV1, TRPV2, and TRPV3 channels[63,64]. It is also possible that the extraordinary, STIM-independent activation of Orai3 by 2-APB[47,65–68] may reflect the aromatic biphenyls in 2-APB mimicking the function of Phe-His pairs in STIM1, and locking the C-terminal M4x peptide in Orai3 into the channel activating configuration[39].

Overall, our studies reveal the critical role of the outer Phe-His pair of residues located on the exposed Sα2 helical apex of STIM1. The Phe-394 residue has a key role in Orai1-binding and in STIM1 is helically aligned with His-398 to facilitate channel gating. The His-398 residue has an indispensable role in transducing STIM1 binding into Orai1 channel gating even though this His residue does not measurably contribute to Orai1-binding. In activated, unfolded STIM1, the Phe-His pair is complexed with Orai1 at the PM surface. In resting, folded STIM1, the Phe-His pair is deeply occluded within the STIM1 CC1 helices, and located very close to the ER membrane surface. The local environment of Phe-394 within folded STIM1 is pivotal, and altered charge at this residue confers either constitutive unclamping or irreversible clamping of STIM1. The decisive roles of these Phe and His residues in the resting and Orai-coupling states of STIM1, will be key to further deciphering the mechanism of STIM1 operation through structural and molecular dynamic approaches.

## Methods

### DNA constructs
STIM1-YFP fragments were derived by PCR from MO91-STIM1-YFP (addgene #19754) and inserted into pEYFP-C1 (Clontech) between XhoI/HindIII sites. The mCherry-STIM1 plasmid was provided by Dr. Madesh Muniswamy (UT Health San Antonio). pIRESneo3-Orai1-CFP and pIRESneo3-Orai1-YFP were described previously[38,47]. Point mutations were generated using the QuikChange Lightning Site-Directed Mutagenesis Kit (Agilent Cat No. 210518). All constructs were confirmed by sequencing before transfection. As shown in numerous previous studies, the C-terminal tag does not alter the ability of STIM1 to interact with or activate the Orai1 channel[33,37,38,50,69].

### Cell culture and transfection
HEK STIM1/STIM2 double knock-out cells (HEK-S1S2DKO)[47] were cultured in high glucose DMEM medium (Mediatech; 10-013-CV)) supplemented with 10% FBS, penicillin and streptomycin (Gemini Bioproducts, CA) at 37 °C with 5% CO₂. HEK STIM1/STIM2 double

knock-out cells stably expressing Orai1-CFP (HEK-S1S2dKO-O1CFP)[39], were cultured in the same medium as above, but supplemented with G418 (100 µg/ml). All transfections were undertaken by electroporation at 180 V, 25 ms in 4 mm cuvettes (Molecular Bio-Products) using the Bio-Rad Gene Pulser Xcell system in OPTI-MEM medium. All experiments commenced 18–24 h after transfection. The parental HEK 293 cell line was from ATCC (CRL-3216). The HEK 293-S1S2DKO (STIM1/STIM2 double KO cell line) and the HEK 293-S1S2DKO-Orai1CFP (HEK 293-S1S2DKO cells stably expressing Orai1-CFP) were derived from the parental HEK 293 cells.

### Photostimulation and image analysis
Fluorescence imaging for optogenetically engineered tools was performed on a Nikon Eclipse Ti-E microscope equipped with an A1R-A1 confocal module with LU-N4 laser sources (argon-ion: 405 and 488 nm; diode: 561 nm). 60× oil or 40× oil objectives were used for high resolution imaging and Ca²⁺ imaging. A built-in 488-nm laser source (1% input) was used as a blur light source. For analysis of Ca²⁺ influx in HeLa cells, the green Ca²⁺ indicator GCaMP6s and mCh-tagged STIM1ct fragment were co-expressed. Both 488-nm and 561-nm laser sources were used to excite GFP and mCherry, respectively, with an interval of 8 s. The collected images were analyzed by the NIS-Elements AR microscope imaging software (Nikon, NIS-element AR version 4.0). 30–60 cells were selected to define regions of interest (ROI) for analyzing time-lapse images of Ca²⁺ influx. Photostimulation was applied using a built-in 488-nm laser source (1% input). All experiments were repeated three times. Data are shown as mean ± SEM.

### Cytosolic Ca²⁺ measurements
Cytosolic Ca²⁺ levels were measured by ratiometric imaging using fura-2 between 18–24 h after transfection as described earlier[70]. Loading of fura-2 and imaging were performed in Ca²⁺-free solution containing (mM): 107 NaCl, 7.2 KCl, 1.2 MgCl₂, 11.5 glucose, 20 Hepes-NaOH, pH 7.2. 1 mM CaCl₂ was added as indicated in experiments. Loading of cells with 2 mM fura-2/AM was for 30 min at room temperature, followed by treatment with fura-2-free solution for a further 30 min. Fluorescence ratio imaging was measured utilizing the Leica DMI 8 fluorescence microscope and Hamamatsu camera ORCA-Fusion controlled by Slidebook 6.0 software (Intelligent Imaging Innovations; Denver, CO). Consecutive excitation at 340 nm ($F_{340}$) and 380 nm ($F_{380}$) was applied every 2 s and emission fluorescence was collected at 505 nm. Intracellular Ca²⁺ levels are shown as $F_{340}/F_{380}$ ratios obtained from groups of >15 single cells per coverslip. All Ca²⁺ imaging experiments were performed at room temperature and representative traces of at least three independent repeats are shown as means ± SEM.

### Enhanced fluorescence image analysis
Cell images were obtained using cells that had remained in culture 18–24 h on coverslips. Images were collected in stacks of 10–20 three-dimensional z-axis image planes at 1-µm steps. Slidebook 6.0 software (Intelligent Imaging Solutions) constrained iterative deconvolution function was used to analyze image stacks and derive enhanced deconvolved images. Deconvolved images shown are mostly from the lowest image levels, close to the coverslip showing ER-PM junctions, and are representative of at least three independent experiments. As in earlier studies[39], enhanced images of the cellular distribution of YFP-tagged SOAR constructs (wild-type, or F394E, F394D, F394R, and F394K mutants) shown in Supplementary Fig. 5. Figures were obtained after brief treatment (10–15 s) with 0.05% trypsin (Cellgro). These images were obtained within 30 min of cell application to cover slips (Fisher Scientific), with cells still remaining in their rounded state, providing a clear delineation of the PM boundary. Pearson's correlation coefficients were derived from multiple cells (as shown) combined from 3 or more independent experiments.

## Förster Resonance Energy Transfer (FRET) measurements

Analysis of FRET was undertaken similarly to that described earlier[38,71]. FRET signals between stably expressed Orai1-YFP and transiently expressed YFP-tagged STIM1 or the mutations of STIM1 described, were obtained using a Leica DMI 8 inverted automated fluorescence microscope equipped with CFP (438Ex/483Em), YFP (500Ex/542Em), and FRET (438Ex/542Em) filter cubes. Images were captured at 20 s intervals to minimize photobleaching. At each time point, 3 sets of images (CFP, YFP and FRET) were collected at room temperature using a 40× oil objective (N.A.1.35; Leica) and processed using Slidebook 6.0 software (Intelligent Imaging Innovations). Images were captured at 20 s intervals. Exposure times for the CFP, YFP and FRET channels were 1000 ms, 250 ms, and 1000 ms, respectively. The decreased YFP channel exposure time compensates for the greater fluorescence intensity of YFP compared to CFP. Three-channel corrected FRET was calculated using the formula:

$$F_C = I_{DA} - Fd/Dd * I_{DD} - Fa/Da * I_{AA} \qquad (1)$$

in which $I_{DD}$, $I_{AA}$ and $I_{DA}$ represent intensity of the background-subtracted CFP, YFP and FRET images, respectively. $F_C$ represents the corrected energy transfer. Fd/Dd represents measured bleed-through of CFP through the FRET filter (0.457), and Fa/Da is measured bleed-through of YFP through the FRET filter (0.190). We used the E-FRET method to analyze 3-cube FRET images as described by Zal and Gascoigne[72] using the formula:

$$E_{app} = F_C / (F_C + G * I_{DD}). \qquad (2)$$

where $G$ is the instrument specific constant. The value of $G$ was determined by measuring the CFP fluorescence increase after YFP acceptor photobleaching using HEK cells transiently transfected with the pEYFP-ECFP construct as described earlier[37]. The value of $G$ was calculated to be $1.64 \pm 0.1$. For all E-FRET summary data and for the FRET images shown, regions of interest were selected close to the plasma membrane around the perimeter of cells as previously described[38,71]. E-FRET analyses were derived from cells with similar YFP/CFP ratios.

## Analysis of puncta

Puncta for STIM1 constructs were analyzed after transient transfection by taking Z-stack images 18 h post-transfection. Z-stacks were taken throughout the entire field of view, before and 5 min after 2.5 μM ionomycin treatment. After image collection, the degree of punctal formation was assessed at depths close to the coverslip, and punctality was recorded for all the positively-transfected cells in the field of view. Cells were considered as punctal if the STIM1-YFP formed puncta with little to no cisternal STIM-YFP fluorescence. Analysis was conducted by counting the number of punctal vs non-punctal cells in 5-7 field of view images with 20–30 cells per field of view, for each STIM1 construct, in each case combining the numbers of counted cells from at least three independent experiments. Punctal size analysis was undertaken using the customized macro plug-in within the ImageJ software, modified from our previously published protocol[73]. Punctal images were converted from 16 to 8 bits, and background was subtracted using a rolling ball with radius larger than 15 pixels. The resulting images were converted to binary images and then processed using the Watershed separation function. Statistics for puncta were obtained with the Analyze particles function in ImageJ, selecting particles with a size between 4 and 300 pixels.

## Structural modeling

The SOAR structure in Fig. 1, Fig. 4a, Fig. 5k, Fig. 7 and Supplementary Fig. 1b, c was derived from the crystal structure[40] (PDB ID: 3TEQ). In Fig. 6l, the structure simulation function of the Chimera software was used to produce the structure of STIM1-CC1α1[74]. The solved solution NMR structure of the STIM1 coiled-coil 1 fragment from STIM1 (PDB ID: 6YEL) was selected as the template[24].

## Statistics

All the data analyses were performed with GraphPad Prism 7 (GraphPad Software). Where shown in figures, Student's t test was used for statistical comparisons between two groups. Data are presented as means ± SEM.

## Reporting summary

Further information on research design is available in the Nature Portfolio Reporting Summary linked to this article.

## Data availability

The data that support this study are available from the corresponding authors upon request. All data supporting the findings of this manuscript are available in the main text or supplementary information. A reporting summary for this article is also available as a Supplementary Information file. Source data are provided with this paper, including source data files for Figs. 1b, c, d, e, g, h, i, j, 2g, h, i, j, 3a, b, d, e, 4c, d, f, g, i, j, l, m, o, p, 5g, h, i, j., and Supplementary Figs. 2a, 3a–h, 4a, c, d, 5f, g, 6a–f, c, d, 10, and 12c, d. Source data are provided with this paper.

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

## Acknowledgements

This work was supported by NIH R35 Grant GM131916 to D.L.G., NIH F31 Predoctoral Fellowship GM125376 to R.M.N., NIH F31 Predoctoral Fellowship HL152619 to J.H.B., a Penn State University Junior Faculty Development Grant to Y.Z., and the Leonard (Jim) Jefferson Endowed Professorship to D.L.G.

## Author contributions

D.L.G. and Y.Z. conceived of the project, designed and undertook experiments, contributed to interpretation of results, and wrote the manuscript. M.R.J. designed and undertook experiments and contributed to interpretation of results and manuscript writing. S.A.K. designed and undertook experiments and contributed to interpretation of results. J.H.B., R.M.N. and Y.W. provided valuable insights into the project, interpretation of results, and assisted in manuscript writing. G.M. and Y.H. undertook experiments, provided reagents and expertise, and assisted in manuscript writing. M.V-R. and M.L.G. undertook experiments and analyzed data.

## Competing interests

The authors declare no competing interests.
