## [Peer Review File · Nature Communications]

An apical Phe-His pair defines the Orai1-coupling site and its occlusion within STIM1Reviewers' Comments:

Reviewer #1:

Remarks to the Author:

In this work by Zhou et al. the role of two sets of paired amino acids in the clustering and activation of Orai1 in STIM1 are explored. The authors find that these two sets of residues function in different aspects of clustering, coupling, and activation. One specific residue- H398-in STIM1 is particularly important for transducing the conformational opening of STIM1 into the allosteric activation of the Orai1 channel. The authors demonstrate these effects through a combination of light microscopy, calcium imaging, mutagenesis, and FRET measurements inside cells. In total, I find the paper fairly straightforward and convincing. It is an important problem for the field. The experiments are well done and easy to follow. I appreciate the use of different complementary systems to investigate this problem. I have several comments and suggestions that could improve the manuscript.

1. Throughout the paper (e.g. Fig. 2,3,4) red/green/overlay images of single cells are shown to illustrate the clustering behavior of the channel mutants. These images are difficult to evaluate. I would recommend that the authors a) switch to magenta/cyan or green color scheme for color-blind readers, and b) provide some extensive quantification of the correlation between the two channels across multiple cells and experiments (pearson correlation for example). As it stands, it is difficult to evaluate the quality or behavior of the channels in these small images.
2. In the abstract the authors state that “-CC1 maps close to the ER surface”. Nowhere in the paper do the authors track the movement of this helix towards or away-from the membrane or the structure of STIM1 relative. The authors FRET data can only support that the two channels associate or rearrange. Please adjust this detail to reflect the information the FRET data can provide and please try to avoid discussions of movement directions (vectors) relative to the membrane or the channel pore without additional experiments to test these ideas.
3. I would appreciate additional discussion of how H398 might interact and activate Ori1.
4. Can the authors provide a sequence alignment of distantly-related STIM1 (or other STIM) isoforms across species to show how well conserved these residues are at these positions as a supplement? This would help the reader in thinking about the SOAR domain, these specific interactions, and whether diseases or other isoforms might behave differently.
5. Please list the PDB number used to generate Fig 1c in the legend.
6. In Figure 2a legend “high-resolution” is ambiguous. It appears to be standard diffraction limited confocal imaging.
7. Please add scale bars to all images or sets of images.
8. Please put number of measurements/cells/biological replicates etc. in the figure legends for all averaged data and bar graphs. I would recommend using a graphing style where the individual points are presented next to (or over) the bar graph so the reader can clearly evaluate the spread of the data.
9. In some graphs the authors use E-FRET and others a normalized E-FRET (normalized to 1). Is there a reason for this change in presentation?

Reviewer #2:

Remarks to the Author:

The manuscript by Zhou et al., describes how two residues in the channel activating domain (CAD or SOAR) of STIM1 define the binding site for activation of Orai1 channels, adding more molecular detail to an already extensively studied interaction.

In Wang et al., Nature Comm 2014 (Figure 6), the Gill group already described in detail the effects of mutating Phenylalanine 394 in the STIM1 activating domain and showed how this mutation in part explains the reduces Orai activation seen with STIM2 (which contains a leucine at this location). Overall, there is rather incremental new knowledge conferred by the current study, although some additional mutations of this region and a more detailed analysis of cluster formation in this study reveal some novel information and the role of the two residues. The back-flipped orientation of the SOAR apex towards the ER membrane in the resting state of STIM1 has already been described by van Drop et al (Elife 2021). The proposed interacting residues on the Orai channel side remain speculative. Experimental evidence for mimicking the effects of F394 by mutation of residues in the M4 helix of Orai1 would improve the study. In addition, protein docking models or MD simulations with WT or mutated CAD domains to the WT Orai1 structure could add further evidence, however, a co-crystallization of both STIM1 (CAD) and Orai1 is ultimately needed.

Figure 1 and 7 are model figures with no experimental data.

Figure 2, 4 and 5: The quality of the fluorescent images is below standard, often signals are saturated. It is impossible to judge clustering of STIM1 or Orai1 as in some images as the plasma membrane is out of focus. No quantification of cluster size and distribution is given. No FRET images are shown. TIRFM or advanced deconvolution should be used to avoid/reduce out-of-focus fluorescence.

Figure 6 needs to include a time series of cluster formation, ideally using TIRFM to reduce out of focus fluorescence. Deletion of only the PBD prevents STIM1 from detaching from MT. What happens if F395K-Delta PBD contains an additional mutation to delete the TRIP motif (see Kim JH et al 2022) for a abbreviated STIM1 GOF?

Images in Supplementary Fig 3 show dead cells. HEK cells are not round.

RESPONSES to REVIEWER COMMENTS

Reviewer #1 (Remarks to the Author):

In this work by Zhou et al. the role of two sets of paired amino acids in the clustering and activation of Orai1 in STIM1 are explored. The authors find that these two sets of residues function in different aspects of clustering, coupling, and activation. One specific residue- H398- in STIM1 is particularly important for transducing the conformational opening of STIM1 into the allosteric activation of the Orai1 channel. The authors demonstrate these effects through a combination of light microscopy, calcium imaging, mutagenesis, and FRET measurements inside cells. In total, I find the paper fairly straightforward and convincing. It is an important problem for the field. The experiments are well done and easy to follow. I appreciate the use of different complementary systems to investigate this problem. I have several comments and suggestions that could improve the manuscript.

We thank the reviewer for a very positive summary of the significance and importance of our work and for recognizing our use of a range of complementary approaches to provide a convincing new perspective in understanding STIM-Orai signaling.

1. Throughout the paper (e.g. Fig. 2,3,4) red/green/overlay images of single cells are shown to illustrate the clustering behavior of the channel mutants. These images are difficult to evaluate. I would recommend that the authors a) switch to magenta/cyan or green color scheme for color-blind readers, and b) provide some extensive quantification of the correlation between the two channels across multiple cells and experiments (pearson correlation for example). As it stands, it is difficult to evaluate the quality or behavior of the channels in these small images.

We carefully considered these comments of the reviewer. As the reviewer suggested, we spent quite some time converting all the red/green overlay images to magenta/cyan. After extensively comparing the red/green vs. magenta/cyan images, we were of the strong opinion that the overlap shown as yellow in the red/green images was much clearer to see than the white overlap in the magenta/cyan images. We were still prepared to go with the magenta/cyan images since, as the reviewer suggested, these might be more visible to red/green colorblind individuals. However, we actually allowed a red/green colorblind colleague to review both sets of images. To our surprise, the colorblind individual actually favored the red/green images. The individual stated that the cyan was difficult to see, and additionally, the individual could see the yellow overlap in the red/green images which was important. After this extensive evaluation, we decided to retain the red/green images. But we much appreciate being able to assess this important point of the reviewer.

We agree with the reviewer that the correlation of overlap of STIM and Orai fluorescence was not so easy to visually assess in a number of the images. As suggested by reviewer 2, we have now replaced almost all the fluorescence images in the figures with those using advanced deconvolution analysis to give more precise definition of the ER-PM junctional areas close the coverslip. This considerably enhances the precision of viewing the near-PM puncta, and removes much of the out of focus fluorescence which made it difficult to evaluate the quality of the images as both reviewers pointed out. However, important to emphasize, our 3-channel FRET data throughout the paper provides enormously greater information than overlap of fluorescence, showing precisely the degree to which the two proteins are actually physically coupled as opposed to the degree of spatial overlap. Indeed, we derive almost all our quantitative measurements on the ability of STIM1 mutants to interact with Orai1 from the E-

FRET data. For example, we reveal that the STIM1-F394K-YFP mutation forms avid and extensive puncta within ER-PM junctions immediately adjacent to the PM. However, there is no visible redistribution of Orai1-CFP into the punctal areas that are defined by STIM1-F394K-YFP (Fig. 5f). The E-FRET measurements (Fig. 5g, and 5i) reveal there is no interaction at all between STIM1-F394K-YFP and Orai1-CFP. We view this as considerably more conclusive than calculating Pearson Coefficients from the fluorescence images. Moreover, the movement of STIM1 into areas adjacent to the PM where Orai1 is already located, can cause spurious positive Pearson numbers even though there is no STIM1-Orai1 interaction.

2. In the abstract the authors state that “-CC1 maps close to the ER surface”. Nowhere in the paper do the authors track the movement of this helix towards or away-from the membrane or the structure of STIM1 relative. The authors FRET data can only support that the two channels associate or rearrange. Please adjust this detail to reflect the information the FRET data can provide and please try to avoid discussions of movement directions (vectors) relative to the membrane or the channel pore without additional experiments to test these ideas.

We agree with the reviewer that nowhere in the paper have we actually “mapped” the distances of the STIM1 structure to the membrane. We, have therefore modified the text to remove such inference to mapping and distances (abstract, line 5; page 5, line 13-15; page 15, line 26-27 page 16, line 17-18, 22-23; Fig. 6 legend title, page 29 and thank the reviewer for allowing us to clarify our conclusions.

3. I would appreciate additional discussion of how H398 might interact and activate Ori1.

The reviewer raises an important discussion point. We know the H398 does not contribute significantly to the overall binding of STIM1 to Orai1, but is crucial for coupling to gate the channel. But we have now added two sentences in the discussion (page 17, last 5 lines), which speculates that the H398 may undergo an aromatic pairing with Orai1, possibly the H264 residue in the “nexus” of Orai1 or possibly the H165 residue in the 2-3 loop of Orai1, both of which we suggested in earlier work, might be important sites through which STIM1 conformationally couples to trigger Orai1 channel gating.

4. Can the authors provide a sequence alignment of distantly-related STIM1 (or other STIM) isoforms across species to show how well conserved these residues are at these positions as a supplement? This would help the reader in thinking about the SOAR domain, these specific interactions, and whether diseases or other isoforms might behave differently.

We much appreciate this suggestion from the reviewer. We have now added a new Supplemental figure comparing the SOAR sequences within STIM1 from human, macaque, mouse, Drosophila, and C. elegans (shown in the new Supplementary Fig. 10). The sequences reveal almost perfect alignment within the mammalian STIM1 proteins, and preservation of the critical apical outer Phe-His pair. Interestingly, the single STIM protein from Drosophila, also has a preserved apical outer Phe-His pair, this despite the fact that the SOAR apex has several other substitutions. Even more interesting, the C. elegans STIM sequence does not have any Phe-His pair in the SOAR apical region. This correlates very well with the fact that recent data from Kim et al (2018) reveal that the C. elegans STIM protein does not interact with or activate mammalian Orai1 channels. The C. elegans STIM protein does interact with the single C. elegans Orai channel, but the crucial TM4 nexus TM4-extension of C elegans Orai are very different to mammalian Orai. We have added text to the paper (page 19, last line; page 20, first 2 lines) to make these interesting points.

5. Please list the PDB number used to generate Fig 1c in the legend.

We have listed the PDB number for the SOAR structure to the Fig. 1c legend. We have also added the PDB number for the CC1 α 1 structure to Fig. 6 and Suppl. Fig. 8. These structures are also listed in the methods under “structural modeling”.

6. In Figure 2a legend “high-resolution” is ambiguous. It appears to be standard diffraction limited confocal imaging.

We have deleted the word “high-resolution” as requested and replaced with “deconvolved images”

7. Please add scale bars to all images or sets of images.

We have now added scale bars to the all images in the figures. We apologize these were omitted in the original manuscript.

8. Please put number of measurements/cells/biological replicates etc. in the figure legends for all averaged data and bar graphs. I would recommend using a graphing style where the individual points are presented next to (or over) the bar graph so the reader can clearly evaluate the spread of the data.

We have now added cell numbers analyzed in all the figure legends. Although the reviewer suggested showing the spread of points, the data presented in the summary bar charts are in almost all cases from large numbers of cells and highly significant with little overlap or any equivocality in interpretation of the data. The number of analyzed cells is much greater than the minimal number (10) required by the journal to show actual point-spread. Hence, there is not a need to add the individual points to show spread.

9. In some graphs the authors use E-FRET and others a normalized E-FRET (normalized to 1). Is there a reason for this change in presentation?

Only in Fig. 3n did we use normalized FRET. This is because there are many possibilities for how the hexamer may be made up of YFP vs. CFP. Thus, when Orai1-CFP and Orai1-YFP are co-expressed, the Orai1 hexamer will form randomly with Orai1-CFP and Orai1-YFP, and it is not possible to control the cells to form a hexamer with a specific stoichiometry. Therefore, the expressed Orai1 hexamer may exist in various possibilities: 6 Orai1-CFP, 1 Orai1-CFP + 5 Orai1-YFP, 2 Orai1-CFP + 4 Orai1-YFP, 3 Orai1-CFP + 3 Orai1-YFP, 2 Orai1-CFP + 4 Orai1-YFP, 1 Orai1-CFP + 5 Orai1-YFP, and 6 Orai1-YFP. Moreover, the percentile of each possibility will vary among different cells, and cannot controlled. As a result, the basal E-FRET value will differ significantly among cells, and the E-FRET value will also vary between different repeats undertaken on different days. Therefore, the importance of this experiment is the change in FRET. The collected E-FRET values were normalized to time 0, which eliminates unwanted background noise. Thus, the error between different cells and different repeats will be minimized. This method was developed by Prakriya in 2008. For all other figures, there are no such concerns, so standard E-FRET values were used.

Reviewer #2 (Remarks to the Author):

The manuscript by Zhou et al., describes how two residues in the channel activating domain

(CAD or SOAR) of STIM1 define the binding site for activation of Orai1 channels, adding more molecular detail to an already extensively studied interaction. In Wang et al., Nature Comm 2014 (Figure 6), the Gill group already described in detail the effects of mutating Phenylalanine 394 in the STIM1 activating domain and showed how this mutation in part explains the reduces Orai activation seen with STIM2 (which contains a leucine at this location). Overall, there is rather incremental new knowledge conferred by the current study, although some additional mutations of this region and a more detailed analysis of cluster formation in this study reveal some novel information and the role of the two residues. The back-flipped orientation of the SOAR apex towards the ER membrane in the resting state of STIM1 has already been described by van Drop et al (Elife 2021). The proposed interacting residues on the Orai channel side remain speculative. Experimental evidence for mimicking the effects of F394 by mutation of residues in the M4 helix of Orai1 would improve the study. In addition, protein docking models or MD simulations with WT or mutated CAD domains to the WT Orai1 structure could add further evidence, however, a co-crystallization of both STIM1 (CAD) and Orai1 is ultimately needed.

We thank the reviewer for recognizing the significance of the work in showing how two residues in the channel activating domain of STIM1 define the binding site for activation of Orai1 channels. Indeed, we would point out that the paper describes an extraordinary and hitherto unknown role for an aromatic pair in the active site of STIM1. The most crucial advance is the revelation that the single His-398 residue plays a fundamental role in transducing binding into Orai channel activation, this despite it having no direct role in Orai-binding. We show that the His-398 residue must be paired with the Phe-394 residue across one turn of the S α 2 helix at the critical SOAR apex of STIM1, and this pairing is essential for the successful transduction event that activates the Orai1 channel. In addition, we reveal that the exact same apical locus is critical to holding SOAR close to the ER membrane. The reviewer is correct that van Dorp's elegant paper had indicated the "back-flipped" molecular topography of the SOAR-Orai1 interaction. The back-flipped model is not a conclusion of our paper, but our data provide critical independent verification of this orientation. This conclusion is ancillary to the major findings of the paper that the single apical locus of STIM1 is critical to both, (a) holding the SOAR in its near-ER folded inactive state, and (b) attaching SOAR to the Orai channel close to the PM. This is carefully depicted in Fig. 7.

The reviewer asks the important question of experimental evidence for the interacting residues of the Orai channel that might mimic the F394 mutations. We apologize for not making better reference to our recent paper (Baraniak et al 2021) that reveals the Leu residues on the C-terminal TM4-extension of Orai 1, do indeed mimic the F394 mutations of SOAR. Thus, our mutation studies with whole Orai and M4x peptides reveal that the Leu-273 and Leu-276 residues, thought previously to hold adjacent M4x peptides in a binding pocket for SOAR, are in fact directly interacting with SOAR. In response to the reviewer's questions, we have tried to clarify reference to this recent Orai work in the current paper (Page 17, line 22-28). Also important in the current study, is that the F394 residue is only part of the binding site on SOAR. Thus, our work in this paper shows that a number of residues in the S α 2 and adjacent helices of SOAR are involved in Orai-binding. But a major point of the paper is to show that "functional coupling" to gate the Orai1 channel is mediated specifically by the His-398 residue held within the Phe-His pair at the SOAR apex.

Given our knowledge of the co-interacting sites in both STIM1 and Orai1, we are in the process of undertaking some advanced molecular dynamics and modeling of this interaction. However, this work is ongoing and requires critical evaluation of the authenticity of the predicted models, and we do not wish to publish speculative or unvalidated models of such interactions. Indeed,

there have been some recent hypothetical models of these interacting sites (ref. 58) that our current data are at odds with. In response to the reviewer's comment, we have made discussion of this point clearer in the paper (Page 19, lines 1-14)

Figure 1 and 7 are model figures with no experimental data.

Although it is a little unusual to have two non-data diagrams, we felt strongly that these diagrams were critical for describing the background and conclusions of the paper. Fig. 1 provides important structural detail that is crucial background for all the studies. Fig. 7, shows our conclusions, that is, the extraordinary unfolding of STIM1, and translocation of the SOAR apex from the S1 α 1 peptide next to the ER surface, to the M4x peptide of the Orai1 channel next to the PM.

Figure 2, 4 and 5: The quality of the fluorescent images is below standard, often signals are saturated. It is impossible to judge clustering of STIM1 or Orai1 as in some images as the plasma membrane is out of focus. No quantification of cluster size and distribution is given. No FRET images are shown. TIRFM or advanced deconvolution should be used to avoid/reduce out-of-focus fluorescence.

The reviewer's point is similar to that raised by Reviewer 1 (Point 1; second part) that we addressed above. We greatly apologize to the reviewer that we did not consistently show advanced deconvolved images for all the cells in these figures. We had used many images in Figs. 2 and 4 that were not deconvolved. The reason for doing this was to show the out-of-focus STIM1-YFP located in the central cellular ER that had not relocated into the ER junctions close to the PM at the bottom of the cell, adjacent to the coverslip. However, as also pointed out by Reviewer 1, we agree with both reviewers that these non-deconvolved images appeared out-of-focus and were more difficult to interpret. We have now been consistent and replaced all the cell images in Figs. 2 and 4 with those that were obtained using advanced deconvolution to avoid out-of-focus fluorescence. We agree that these images do greatly improve visualization of the movement of STIM1-YFP and Orai1-CFP into puncta, and also allow us to better show the lack of movement of Orai1 when the interaction with STIM1 is prevented by the critical STIM1 mutations. We greatly appreciate the reviewers pointing this out, and the new images considerably enhance the paper. The clustering observed is now very clear and we see no need for quantification of cluster size or presentation of FRET images. We should point out that in Fig. 5, we did retain some non-deconvolved images since we were showing cell images before and after store-depletion, and it was important to visualize the loss of STIM1 from central ER in the cell body, and see it appear as puncta as the STIM1 moves into ER-PM junctions, in those cells in which STIM1 movement was activated by store-depletion.

Figure 6 needs to include a time series of cluster formation, ideally using TIRFM to reduce out of focus fluorescence.

As stated in response to the previous point, we have used deconvolved images not TIRFM to reduce out-of-focus fluorescence. With regard to time-series for puncta formation, in Figs. 6a to 6g, all the images shown are either constitutively punctal STIM1 that does not change before or after store-depletion, or constitutively non-punctal STIM1 that remains non-punctal after store-depletion. We believe that time series of images would not provide more useful information. In Fig. 6h – 6n, we are showing only the distribution of STIM1 in resting, store-replete cells, and not showing any time-related changes in STIM1 location.

Deletion of only the PBD prevents STIM1 from detaching from MT. What happens if F395K-Delta PBD contains an additional mutation to delete the TRIP motif (see Kim JH et al 2022) for an abbreviated STIM1 GOF?

The reviewer raises an interesting question that we had not addressed concerning the relationship between the F394K mutation (reference to “F395K” was a typo) described in our paper here, and the interesting effects of altering the TRIP motif (the STIM1 642-645 sequence), as undertaken in the recent paper by Kim et al (2022). The TRIP motif is an EB-1 binding site that can stabilize the inactive folded state of STIM1 by enhancing its interaction with microtubules in the vicinity of the ER. Thus, elimination of the TRIP motif promotes the ability of STIM1 to translocate into ER-PM junctions.

As requested by the reviewer, we tested the construct in which the STIM1-F394K mutation was combined with the “Delta-PBD” (we call this Δ polyK), and compared this construct with one containing the same two mutations together with the STIM1 (642-645) “TRNN” mutation to eliminate the TRIP motif (data are shown in the new Supplementary Fig. 6; described in the text on page 13, lines 18-22). We had shown in the original paper, that the Δ polyK mutation (removing the C-terminal poly-K region of STIM1), completely prevents the ability of the constitutively punctal STIM1-F394K mutation to relocate into puncta. As shown in Supplementary Fig. 6, our new data reveal that the TRNN does not rescue the defect caused by the Δ polyK that prevents the F394K mutation-driven movement into puncta. This is important and reveals that the GoF TRNN mutation that causes constitutive activation of STIM1 cannot overcome the lack of attachment of STIM1-F394K to the PM as a result of removing the poly-K domain. It should be noted, however, that the enhancing effect of the TRIP-deletion shown by Kim et al (2022), was at least partially dependent on the ability of STIM1 to bind to Orai1 channels. Thus, the inability of the STIM1-F394K mutation to interact with Orai channels may have contributed to the lack of effect of the TRIP deletion.

Images in Supplementary Fig 3 show dead cells. HEK cells are not round.

We apologize to the reviewer, that we omitted to explain our cell imaging deconvolution procedures, and also, we omitted an explanation of the use of recently plated, rounded cells in order to provide a clear delineation of the PM boundary. We have used this latter technique to obtain deconvolved images of cytosolically-expressed SOAR in several recent papers. Focusing on the mid-level of the cells, this allows us to clearly observe SOAR attachment to the PM. We have now added a new section under Methods entitled “Enhanced Fluorescence Image Analysis” which describes this technique in particular, and our enhanced image analysis in general (Page 23, middle para.).

Reviewers' Comments:

Reviewer #1:

Remarks to the Author:

The authors have addressed my concerns from the first version of the manuscript. The manuscript is now improved. I have no further comments.

Reviewer #2:

Remarks to the Author:

The authors have addressed several of the previously raised concerns in their rebuttal letter, especially regarding the interpretation of their previously published data regarding the role of Phenylalanine 394, which is not conserved in the close STIM1 homolog STIM2. However, they need to clarify this homolog discrepancy more clearly within their manuscript abstract and introductory text as well. The abstract states that the "F394-H398 pair is the crucial Orai1 coupling locus", with "Phe394 critical for binding Orai1", which is misleading given that STIM2 (containing a Leucine at this position) by itself can activate Orai1, although less efficiently. Although this is discussed later (line 456 ff), the wording in the abstract and introduction gives the impression that Phe394 is absolutely required, which it clearly is not. Why was the F394L mutation (as published by the authors previously) not included in Figure 4? The authors claim that many previous studies use only short fragments of the STIM1 protein, however, they also need to discuss their use of constructs with a large C-terminal YFP tag, which reduces STIM-ORAI functionality and occludes the STIM1 C-terminal polybasic domain compared to constructs in which the YFP tag is placed upstream of the PBD.

Deconvolution analysis has improved image quality, however, TIRFM would have still been the method of choice. As mentioned in the previous concern, it remains unclear why the authors did not show the FRET images to correlate with the E-FRET analysis. This is likely because the E-FRET differences are small. Do constructs in which the YFP tag is localized immediately upstream of the PBD show increased FRET? Shown in Fig 2a and f are colocalization images, which would require colocalization analysis using Pearson's or Mander's coefficients, which the authors argue against in their reply to reviewer 1, but then E-FRET images should be shown. The authors have also not properly addressed why cluster sizes were not quantified, this is a relevant point as they quantify the percentage of cells with puncta in Figure 6, but it is unclear which size defines a puncta.

Minor additional point: the error bars in Figs. 2e are not symmetrical. Why are not all error bars shown as two-sided error bars?

I strongly disagree with having two model figures in the main manuscript. Figure 1 can be a supplementary figure or essential parts (apical SOAR) can be part of figure 2.

The discussion of the second model (Figure 7) needs to include the authors previous models of STIM1 unimolecular coupling to ORAI channels (Zhou et al., Nature comm 2015).

The new Supplementary Figure 10 should include the STIM2 sequence.

The reviewer appreciates the additional analysis of the EB binding site mutant (Fig. S6).

Overall, the study is entirely performed using tagged and overexpressed constructs in heterologous overexpression which should be critically discussed.

AUTHORS RESPONSES TO REVIEWER COMMENTS (09-08-2023)

Response to Reviewer #1

The authors have addressed my concerns from the first version of the manuscript. The manuscript is now improved. I have no further comments.

We are glad that the concerns of Reviewer #1 were satisfactorily addressed and we thank the reviewer for their inciteful and most helpful feedback that allowed us to significantly improve the manuscript.

Reponse to Reviewer #2

The authors have addressed several of the previously raised concerns in their rebuttal letter, especially regarding the interpretation of their previously published data regarding the role of Phenylalanine 394, which is not conserved in the close STIM1 homolog STIM2.

We thank Reviewer #2 for providing important suggestions and we are glad that the reviewer recognized we have addressed many of the concerns that were raised.

However, they need to clarify this homolog discrepancy more clearly within their manuscript abstract and introductory text as well. The abstract states that the “F394-H398 pair is the crucial Orai1 coupling locus”, with “Phe394 critical for binding Orai1”, which is misleading given that STIM2 (containing a Leucine at this position) by itself can activate Orai1, although less efficiently. Although this is discussed later (line 456 ff), the wording in the abstract and introduction gives the impression that Phe394 is absolutely required, which it clearly is not. Why was the F394L mutation (as published by the authors previously) not included in Figure 4?

The reviewer raises a further important point and we apologize that the abstract and introduction gave the impression that the F394 residues is absolutely required for coupling to Orai1, whereas in STIM2 this F394 is a Leu residue. We have modified the wording in both the Abstract and Introduction to indicate that the F394 residue is important (but not absolutely required) in mediating binding in STIM1 (Page 2, lines 4, and 8-9; Page 5, lines 2-3). As the reviewer points out, we have discussed this more fully in the Discussion and pointed out that this substantial difference between STIM1 and STIM2 in coupling to Orai1 channels, likely explains the lower efficacy of Orai1 channel activation and the greater degree of constitutive activation of STIM2 compared to STIM1. We have now clarified this discussion better (Page 20, lines 8-9 from bottom). The F394L data was extensively published in our previous paper (Wang et al 2014, Nature Comm) so we could not include repeats of these data in the current paper. However, the current paper provides an important explanation for why the lack of the Phe-His pair in STIM2 is likely the cause for the major differences in the function of STIM2 compared to STIM1.

The authors claim that many previous studies use only short fragments of the STIM1 protein, however, they also need to discuss their use of constructs with a large C-terminal YFP tag, which reduces STIM-ORAI functionality and occludes the STIM1 C-terminal polybasic domain compared to constructs in which the YFP tag is placed upstream of the PBD.

The reviewer raises an important question about the tagged constructs we used and again we apologize that we had not made this clear in the previous version of the paper. We have added

a small section in the Methods (Page 22, lines 9-11) including referenced papers, to describe that the C-terminal YFP tag does not affect functional coupling to Orai1, and has been used by numerous investigators in a great many papers (including our own) to mediate STIM1 activation. We now include reference to the study by Muik et al (2008) showing there are no significant differences between, N-terminally vs. C-terminally tagged STIM1, in functional coupling to Orai1.

A further interesting question raised by the reviewer is whether the small poly-basic domain (PBD; residues 671-685) at the far C-terminus of STIM1, would still interact with the acidic lipids in the PM despite having the immediately adjacent YFP tag. Indeed, we realized that the data in the current paper, gives an exact answer to Reviewer 2's question, and we have added a sentence in the paper to describe this important information (Page 13, lines 20-22). Thus, our results in Fig. 5 (was Fig. 6), reveal that the powerful F394K mutation of STIM1-YFP, which is completely unable to interact with Orai1, still avidly binds to the PM lipids through its poly-K region, despite having the C-terminal YFP tag placed immediately adjacent to the poly-K region. We show that removal of the poly-K region completely blocks the STIM1-F394K-YFP construct from interacting with the PM. We thank the reviewer for asking this question which allows us to make this important point.

Deconvolution analysis has improved image quality, however, TIRFM would have still been the method of choice. As mentioned in the previous concern, it remains unclear why the authors did not show the FRET images to correlate with the E-FRET analysis. This is likely because the E-FRET differences are small. Do constructs in which the YFP tag is localized immediately upstream of the PBD show increased FRET? Shown in Fig 2a and f are colocalization images, which would require colocalization analysis using Pearson's or Mander's coefficients, which the authors argue against in their reply to reviewer 1, but then E-FRET images should be shown.

We are glad that the Reviewer recognized that our replacement of all images with deconvolved images, substantially improved the quality of our images, and that the TIRFM analyses are not necessary. We apologize that we did not go further with the Reviewer's earlier requests that we provide FRET images and undertake Pearson's coefficient analysis. We have now done exactly as the Reviewer requested on both counts. Thus, we have now added a new Supplementary figure (now Suppl. Fig. 2b-h) in which we provide all the FRET images for the FRET data shown in Fig. 1 (was Fig. 2), as requested. This Suppl. Fig. 2, is now referred to in the text (Page 6, lines 4 and 5). We agree, that these are useful and extremely clear, and provide complementary visual information to the E-FRET data shown. We have also added some wording to the FRET methodology to include reference to the FRET imaging procedure (Page 24, lines 19-22). The E-FRET data and images show large, robust changes, not small changes as the reviewer speculated. Also, as explained and referenced above, the YFP C-terminal tag, does not affect the interaction between STIM1 and Orai1.

Again, as the reviewer requested, we have now added Pearson's coefficient measurements for the co-localization images shown in Fig. 1a and f (was Fig. 2). These data are shown in the new Suppl. Fig. 2a, and described in the new legend. Reference to this analysis is on Page 6, lines 4 and 5, and in Methods, Page 23, lines 4-5 from bottom. We apologize for arguing against the necessity for the Pearson analysis that was requested by the other reviewer (Reviewer 1), and we agree that these data do indeed confirm the colocalization and FRET images. This is very helpful for the reader and we thank Reviewer 2 for allowing us to improve the paper by providing these analyses.

The authors have also not properly addressed why cluster sizes were not quantified, this is a relevant point as they quantify the percentage of cells with puncta in Figure 6, but it is unclear which size defines a puncta.

We apologize that we did not provide the data on cluster sizes for puncta. We have now added another new supplementary figure (Suppl. Fig. 10) in which we show the punctal size distribution for each of the major STIM1 mutants from Fig. 5a-g (was Fig. 6). We refer to this new supplementary data in the text (Page 10, lines 8-10 from bottom; Page 14, lines 1, 21 and 29). The methodology for punctal size measurements is provided on Page 24 lines 1-3 from bottom, and Page 25, lines 1-4.

Minor additional point: the error bars in Figs. 2e are not symmetrical. Why are not all error bars shown as two-sided error bars?

We have changed all the error bars to 2-sided error bars in bar charts. All drawing programs use the top of the box line as the value. When box lines are thicker and errors are very small, it makes the error bars look unsymmetrical. We made this visually better by making the box lines thinner.

I strongly disagree with having two model figures in the main manuscript. Figure 1 can be a supplementary figure or essentials parts (apical SOAR) can be part of figure 2.

We have reluctantly complied with the reviewer on this. Former Fig 1 was crucial to the whole introduction and earlier part of the paper and, as in many of our major papers, we like to set the stage visually with a graphical fig in the main text. However, it is now Suppl. Fig. 1.

The discussion of the second model (Figure 7) needs to include the authors previous models of STIM1 unimolecular coupling to ORAI channels (Zhou et al., Nature comm 2015).

We have now included the reference to our unimolecular coupling model for STIM1 published in our earlier Nature Communications paper in the discussion of the new model which is now Fig. 6. (Page 18, lines 4-8)

The new Supplementary Figure 10 should include the STIM2 sequence.

This was a great idea and we added the sequence to Suppl. Fig.13 (was Suppl. Fig. 10)

The reviewer appreciates the additional analysis of the EB binding site mutant (Fig. S6).

This was an interesting possibility and we dedicated much effort to conducting these studies.

Overall, the study is entirely performed using tagged and overexpressed constructs in heterologous overexpression which should be critically discussed.

We do not understand the basis of this comment since our work is all based on homologous expression of single proteins within clean CRISPR-derived knockout backgrounds, completely devoid of endogenous proteins. We consider that this is one of the profound strengths of our paper.